# Temperature difference and systemic risk: Evidence from LASSO-VAR-DY based on China's pan-financial market

**Kaiwei Jia**[ID]*[☯], **Yunqing Du**[ID][☯]

School of Business Administration, Liaoning Technical University, Huludao, Liaoning, China

☯ These authors contributed equally to this work.
* jiakaiwei@lntu.edu.cn

**Data Availability Statement:** The data underlying the results presented in the study are available from Wind, CSMAR and ECMWF databases.The minimum data set used in this article can be found

## Abstract

Climate change-induced pan-financial market and the contagion of systemic financial risks are becoming important issues in the financial sector. The paper measures the temperature difference in terms of the degree and direction of deviation of the actual temperature relative to the average temperature of the same historical period. Based on the high-dimensional time-series variable LASSO-VAR-DY framework, we construct a pan-financial market volatility correlation network consisting of 112 Chinese listed companies in banking, insurance, securities, real estate, traditional energy, and new energy, use eigenvector centrality to measure the systematic risk of each firm, and then empirically test the effect of temperature difference on systematic risk under pan-financial market scenario. The results of the study show that (i) There is a significant difference among the systemic risk of financial sectors such as banking, insurance, and securities in the financial market pan-financial market scenario and the systemic risk when the financial market pan-financial market is not taken into account;(ii) Higher temperature significantly exacerbates systemic financial risk, while colder temperature significantly mitigates systemic risk, but both have an asymmetric effect on systemic risk, and there is sectoral heterogeneity.(iii) From the dynamic evolutionary characteristics, there are significant differences in the response of systemic financial risk to positive and negative temperature shocks;(iv) The results of the systemic risk variance decomposition indicate that the temperature change contributes more to the variance of systemic risk in the banking and securities sectors in pan-financial market;(v) The contagion source of financial systemic risk shows an obvious path of leaping and changing characteristics, and the contagion source of systemic risk (source of impact) shows the evolution law of "bank → real estate → new energy → temperature difference," which means that the temperature difference has become the contagion source of systemic financial risk. This study provides a reference for preventing and resolving systemic risks under pan-financial market scenario and provides a basis for improving the current macroprudential regulatory framework.

in the supporting information section of the Appendix data file.

**Funding:** In 2023, Liaoning Technical University's school-level social science class unveiled the project of China-style bank governance modernization and systemic risk prevention " (No. 23-A017).The funders had no role in study design, data collection and analysis, decision to publish, or preparation of the manuscript.

**Competing interests:** The authors have declared that no competing interests exist.

## Introduction

Climate change refers to changes in the long-term climate state, most notably characterized by an increase in the global average temperature [1]. China is one of the countries most affected by global climate change. Climate change has broad and far-reaching implications for the financial system. Temperature changes have exacerbated economic policy uncertainty, and financial generalization caused by the financialization of the real economy, especially the high-carbon sector, has become increasingly prominent [2–6]. The financialization of high-carbon sectors, such as energy and real estate, is prominent [3,7]. Xie et al. [4] defined nine submarkets in traditional financial market, real estate, and commodity market as pan-financial market. They investigated the extreme risk spillovers and evolutionary patterns in pan-financial market. The results show significant extreme risk spillovers among the submarkets in pan-financial market, and the energy submarket is always the primary recipient of upside extreme risk spillovers. This suggests that energy market plays an essential role in forming and contagion systemic financial risks. Existing research suggests that extreme weather severely affects firms' production, operations, and valuation, weakening the balance sheets of firms and households and tightening bank credit, which exacerbates correlations among financial institutions. Under the financial accelerator effect, the complex web of market signals and financial systems could deepen the severity of climate risks, leading to a protracted tightening of financial conditions, possibly even a "Minsky moment," which could eventually evolve into a systemic risk for the traditional financial sector. Against this backdrop, the Central Committee of the Communist Party of China (CPC) has made a significant decision to respond to global warming, which means that it will "strive to achieve carbon peaking by 2030 and carbon neutrality by 2060", which will lead to extensive and profound economic and social systemic changes, and global climate change as well as institutional and policy arrangements to address extreme climate change are bound to impact China's High-carbon sectors. Based on the above analysis, we believe that it is necessary to re-examine the impact of temperature difference on systemic financial risk: first, the financial generalization problem in the context of temperature difference has become increasingly prominent, the boundaries of the financial sector have gradually blurred and expanded, and the financial system has gradually generalized from the traditional sectors of banking, insurance, and securities to a pan-financial sector that encompasses banking, insurance, securities, real estate, traditional energy, and new and emerging energy sources [3,4]; secondly, due to the expansion of the number of submarkets in the financial system, the path of systemic risk contagion caused by temperature difference changes will become more complex. On the contrary, few scholars have conducted in-depth and systematic investigations on the systemic risk of temperature change on pan-financial market (including the high-carbon sectors where the financialization phenomenon is very prominent, such as the traditional energy, new energy, and real estate sectors) in the existing researches.

The rest of the paper is structured as follows: Part 2 is the literature review; Part 3 is the research design; Part 4 is the measurement and analysis of temperature difference and pan-financial market systemic risk; Part 5 is the empirical analysis of the influence mechanism of pan-financial market systemic risk; Part 6 is the dynamic evolution law and analysis of temperature difference and pan-financial market systemic risk; Part 7 is the temperature difference and pan-financial market systemic risk transmission potential path identification; Part 8 summarizes the whole paper and provides corresponding policy recommendations.

## Literature review

### Literature review of the impact of climate change on financial systemic risk

Climate change means significant and long-lasting changes in the Earth's climate and weather patterns or global warming. It measures the average weather and weather variability at a

particular location, specifically the average state and variability of climatic characteristics, such as temperature, precipitation, humidity, and wind, over a long period [8]. Climate change brings serious environmental problems and potential economic and financial crises. Since Nordhaus's pioneering work on climate economics, a growing body of literature has begun to focus on climate change's economic and financial impacts. It has identified climate change as a critical economic growth variable [9]. Capasso et al. [10] even directly pointed out that the "tragedy of the horizon" caused by climate change has exacerbated financial vulnerability, and the possibility of a "Minsky moment" has risen sharply.

Climate change has been recognized as an emerging source of risk in the financial system, and the relationship between climate change and financial stability has attracted much attention from academics and practitioners [11]. However, significant gaps still need to be found in the exciting and challenging research area of climate finance. One strand of the literature focuses on the impact of climate change on financial stability using theoretical models [12]. Another category focuses on empirical research on the impact of climate change on the financial system, where scholars argue that frequent extreme weather disasters can lead to the loss of physical assets, jeopardize the safety of life and health, and then transmit to the financial sector through the impacts on the real economy, with adverse risk shocks to the financial sector, and the consequent increase in the number of insurance claims, and credit risk exposures, which can jeopardize financial stability [13]. Faccini et al. [14], for the first time, investigated whether U.S. stock prices reflect market-wide climate change risk by constructing four new textual forms of market-wide climate change risk indicators (natural disasters, global warming, international summits, and U.S. climate policy) through textual and narrative analyses of Reuters climate change news. The study found that only climate policy factors were priced, especially after 2012. Furthermore, investors only care about climate change risk when policymakers intervene, and the discussion enters the political arena. Dong & Liu [15] investigated the impact of climate risk on stock prices, analyzing a sample of U.S. firms and finding that firms located in disaster-prone counties are associated with a higher risk of stock market crashes. Further, considering the impact of the energy transition due to the response to climate change, on the one hand, it leads to a rise in stranded assets in the high-carbon sector, including the traditional energy sector, creating non-performing assets in the high-carbon sector and the financial sector, and exacerbating financial instability; on the other hand, the profit-seeking instinct, while driving the deconcentration of the high-carbon sector, objectively creates a cohort effect of the green transition in the high-carbon sector. The cohort effect increases the uncertainty of common risk exposure and transition, and financial market volatility rises significantly, affecting the financial system's stability [16].

There are relatively few empirical studies on the impact of climate change on pan-financial system. Most current scholars study climate change's impact on a single industry or market, such as banking, stocks, bonds, commodities and foreign exchange markets, real estate, and energy assets [17–19]. As for the study of pan-financial systems composed of multiple markets, scholars are currently using the network approach to study the risk contagion among nine sub-markets in traditional financial market, real estate, and commodities market [4], as well as studying the systemic risk spillover effects among China's oil, gold, real estate, and financial sectors, or studying the risk dependence between the banks and the energy sector, the development of the low-carbon transition has led to a more pronounced risk dependence of the banking sector on the new energy sector than on the traditional energy sector [20], but is limited to the study of risk contagion between pan-financial sectors, and there are also scholars who have introduced temperature difference as an exogenous shock and used a network approach to study the impact on the systemic risk of macro-financials (equities, commodities, currencies, and bonds), and have found that climate risk not only affects a single financial market, but also

triggers a synergistic movement of risk, exacerbating potential systemic financial risks. Flori et al. [21] argued that climate risk negatively affects the financial sector, leading to macroeconomic downturns and that the subsectors in the financial system are closely interconnected and interact with each other, further increasing the risk exposures of the financial sector.

## Literature review of the impact of internal factors, external shocks and financial market factors on systemic risk

In addition to climate change, there are also internal factors within institutions, external shocks, and the degree of financial market development that can have a significant impact on systemic risk. On the one hand, part of the literature attempts to explore the micro influences on systemic risk in the context of the structural characteristics of financial institutions. Existing scholars have empirically studied the impact of systemic risk mainly from the perspectives of financial institutions' scales, liquidity, leverage, loan structure, and the organizational and geographic complexity in which the firms are embedded [22,23]. On the other hand, the impact of systemic risk has been studied from the perspective of internal corporate governance, bank regulation, and banks' risk-taking behavior based on investor protection, and other scholars have examined the impact of systemic risk from the perspective of corporate governance. Some scholars study corporate governance's impact on financial institutions' stability based on the perspective of executive compensation incentives and find that there is a herd effect among executives, which triggers institutional affiliation investment and drives risk accumulation [24]. Some scholars study the impact of corporate governance on financial performance based on the ESG perspective.

Many scholars study the impact of external shocks on markets, industries, and enterprises. First, Dicks et al. [25] proposed the theory of investors' "uncertainty aversion" preference and found that uncertainty drives new systemic risk contagion. Economic policy uncertainty has risen sharply recently due to trade frictions, geopolitical risks, and energy crises, making it possible for adverse shocks to spread more broadly throughout the financial system. Guo et al. [26] and Liu et al. [27] studied risk contagion among international stock markets during the COVID-19 pandemic. They found that the COVID-19 epidemic significantly increased the risk contagion effect of international stock markets. However, these studies are based on the impact of unexpected shocks on the Chinese financial sector in the short term. When COVID-19 and hurricanes are compounded, a nonlinear dynamic of amplified losses emerges, negatively affecting economic recovery, bank financial stability, and public debt sustainability [28]. Secondly, Jebabli et al. [29] also study inter-market risk spillovers from a financial crisis perspective, comparatively analyzing financial systemic risk spillovers during multiple financial crisis periods. The impact on default risk has also been studied from the perspective of economic policy uncertainty. Third, there are also some scholars from the perspective of government intervention to study the impact on systemic risk. Many studies have also begun to discuss the moral hazard of government bailouts, market pricing distortions, damage to sovereign credit, and other issues discussed. Based on further research on the impact of government intervention on systemic risk, the main influencing factors are the local government debt, the Chinese government's financial system management model, government assistance, and monetary policy [30,31]. Fourth, shocks in the international trade environment significantly impact systemic risk [32]. With the deepening of the international trade division of labor, economies cooperate, which can effectively mitigate the shock of systemic risk. However, other scholars have shown that with the cooperation of inter-industry trade, the international industrial chain may become a new conduction channel for the spread of financial systemic risk [33].

The level of financial system development significantly affects systemic risk. Economies with higher levels of financial development are conducive to the stability of financial market.

However, some scholars have shown that developed markets produce higher risk spillover effects, adversely affecting the financial sector's stability [34]. At the same time, cross-border flows and the degree of capital openness also significantly impact the systemic risk of financial market. The operating conditions of the macroeconomy also have a significant impact on systemic risk. Studies have shown that when the overall macroeconomic trend is positive, by the value of collateral, as it rises, credit supply tends to be loose, and asset bubbles are significantly raised. With the deterioration of the economy and a large amount of capital recovery, resulting in the bursting of the bubble, many financial institutions began to sell assets, leading to systemic risk in the financial market [35].

## Literature review of systemic risk contagion from a network perspective

Under the financial accelerator effect, the complex financial system network may deepen the severity of climate risk, leading to a lasting tightening of financial conditions, which may eventually evolve into systemic financial risk. In terms of the characteristics of the data on which the construction of the complex network depends, the complex network of the financial system can be categorized into three types: the first type of research utilizes actual bank asset-liability correlation data to construct an inter-institutional network, which can accurately capture the generation and contagion mechanism of systemic risk. However, financial data has the disadvantages of low frequency, high lag, and limited forms of association that can be reflected, making this type of research unable to accurately portray the high-dimensional, time-varying network of associations among institutions; the second type of research is to expand along the social relations of directors and shareholders of financial institutions, focusing on the analysis of financial institutions' main body of business activities, and the risk contagion triggered by the economic behavior in the network [36]. Chen et al.[37] construct a mutual interlocking directors network based on the data of Chinese A-share listed companies, and find that the higher the degree and betweenness centrality of a firm in the interlocking directorate network, the more substantial impact on firm performance. It shows that network topology characteristics significantly impact business operations, which is helpful for selecting variables for the degree of centrality in this paper.

The third type of research mainly uses high-frequency financial market data to study multiple channels of risk transmission, and high-frequency data not only overcomes the shortcomings mentioned above of financial data but also breaks down the barriers caused by the differences in the subjects and calibers of financial data [38,39]. In order to make the correlation information that the network can contain more complete, Demirer et al. [40] refined the network construction method by combining the LASSO estimation method with the VAR model and utilizing the LASSO-VAR-DY framework to construct a high-dimensional correlation network among cross-border banks. At present, this type of research has been widely recognized by domestic academics. A large number of scholars have utilized the average stock price [38], stock price return [41], stock price volatility [42], and the tail risk dependence indicators [39] to construct the correlation network among financial sectors or financial institutions in China. These provide ideas and methods for this paper to analyze the dynamic evolution trend (minimum spanning tree) between sectors based on the connectivity of the stock market, which can better solve the problem of high-dimensional data. At the same time, the connectivity of the stock market also reflects the information efficiency of the capital market, which is conducive to the study of internal contagion and spillover effects of financial risk.

## Review of the literature review

Taking an overview of the existing relevant literature, first of all, as far as the research object is concerned, most of the existing papers study the impact of climate change on the systemic risk

of a single market or the systemic risk of traditional financial market, but few scholars have studied the impact of climate change on the systemic risk of China's traditional financial sector as well as pan-financial market that include pan-financial market including the pan-real estate, traditional energy, and new energy sectors. This paper tries to take pan-financial market of the largest developing countries as a scenario to conduct a quantitative study. Financial market of the largest developing countries as a scenario for quantitative research. Second, in terms of research methodology, most of the existing studies use tail risk dependence indicators, which have apparent shortcomings compared with stock price return and volatility indicators, such as the conditional value-at-risk (CoVaR), marginal expected loss (MES), systematic expected loss (SES), and systemic financial risk index (SRISK) require unique crisis settings or financial assumptions. Compared with stock returns, stock price volatility measures the uncertainty of institutional stock movements, which can reflect investor panic and the market risk level of institutions in different periods and better reflect people's sensitivity to the systemic risk caused by climate change [42,43].Meanwhile, the number of variables involved in pan-financial market risk spillover effect is large, and the dimensionality catastrophe can be overcome using the LASSO method. Therefore, this paper utilizes the lasso algorithm to construct a directed weighted correlation network of pan-financial market stock price volatility. Finally, in terms of research content, there have been abundant studies on the impact of different external shocks on financial market. However, there needs to be more research on the impact of temperature difference and two-way volatility of temperature difference on China's pan-financial market. This paper tries to extend the understanding of temperature difference to study the asymmetric effect of bidirectional fluctuation of temperature difference on systemic risk. It reveals the evolution and leapfrog path of systemic financial risk contagion sources.

In summary, the thesis extends the understanding of temperature difference to measure the bidirectional fluctuation of temperature difference in terms of the degree and direction of deviation of actual temperature relative to the average temperature of the same period in history and draws on the research framework of Demirer et al. [40] to construct the volatility correlation network composed of 112 Chinese banks, insurance, securities, real estate, traditional energy, and new energy listed companies in pan-financial market, using eigenvector centrality to measure the systemic risk of each company, and then empirically examining the effect of temperature difference on systemic risk under pan-financial market scenario, as a way to explore the impact of climate change on the stability of the institutions within each sector of pan-financial system in China.

Compared with existing studies, the first contribution of this paper is to take pan-financial market of the largest developing countries as a scenario to study the mechanism of the impact of temperature difference on systemic financial risk and fill the gap in the mechanism of systemic financial risk formation or provides a new perspective. Although some scholars consider constructing a pan-financial market in China [3,4], these scholars only analyze the risk spillover effects based on pan-financial market at the industry level. Hence, this paper draws on the analytical methodology of Wang et al. [41] analysis of risk spillover effects based on the institutional level to study pan-financial market risk spillover effects in China's impacts, taking into account the fact that the energy and real estate sectors have gradually become pan-financialized with both commodity and actual attributes, driven by a combination of factors such as climate change, financial market structural transformation and changes in regulatory concepts. A China's pan-financial market comprising real estate, energy sector, and traditional financial sector is constructed, and a network analysis is conducted at the firm level to explore the generation and transmission mechanism of systemic risk in the China's pan-financial market.

The second contribution of this paper is to extend the understanding of temperature difference and to emphasize the asymmetric effects of two-way fluctuations in temperature difference. Although previous scholars have defined temperature change as the average day-to-day

temperature [44], and the method of using day-by-day observations of meteorological stations during the baseline period and percentile relative thresholds to define the extreme thresholds of meteorological indicators in different cities [45], the observational methods of these are at an independent level, and there is no comparison of the temperature with the historical temperature Considering the long-term changes in temperature, this paper draws on the method used by Song & Fang [46] to calculate the temperature difference from the moving average of the quarterly temperature deviation from the base period, measures the temperature difference based on the degree and direction of deviation of the actual temperature relative to the average temperature of the same period in the past, and decomposes the temperature difference into the increase and decrease of temperature, to explore the effect of the bi-directional fluctuation of the temperature difference on the asymmetry of the China's pan financial market with asymmetric effects.

The third contribution is to reveal the evolution and leaping path of systemic financial risk contagion sources, which provides a basis for preventing systemic financial risks and improving macroprudential regulatory policies in China and globally. One category of literature on climate change as a new source of systemic risk contagion focuses on the impact of climate change on financial stability using theoretical models [12]. Another category of empirical studies on the impact of climate change on pan-financial system is relatively few and mainly examines the impact of climate change on a single industry or market [11,12,17,18]. In contrast, studies on pan-financial systems composed of multiple markets have been limited to studying risk contagion among pan-financial market [4]. In contrast, this paper starts from the perspective of climate change, constructs a panel vector autoregressive model (PVAR), and conducts an impulse response (IRF) analysis to study the dynamic evolution mechanism of temperature rise and temperature fall on the systemic risk of pan-financial market, and utilizes the minimum spanning tree (MST) algorithm to capture the leaping path of the systemic risk contagion within pan-financial market network, which fills in the gaps of the existing research, and provides an excellent opportunity for the prevention and early warning of systemic financial risk in China and even the world. This study fills the gaps in existing research and provides a basis for decision-making for the prevention and early warning of systemic financial risks in China and globally.

## Research design

### Construction of a pan-financial market volatility spillover network based on the LASSO-VAR-DY framework

Diebold and Yilmaz [42] state that correlation is at the heart of modern risk measurement. Once a financial firm's stock is volatile, it is highly likely to be contagious to other firms. To verify this correlation among firms within pan-financial market, this paper draws on the methodology of Demirer et al.[40], which employs LASSO-VAR and the generalized variance decomposition of Diebold and Yilmaz [42] (LASSO-VAR-DY framework) for network construction on high-dimensional data and maps it into a network topology through variance decomposition, looks at volatility correlations among pan-financial market firms and measures the spillover effect of systemic risk, filters the network using a threshold pair of net pairs of spillover networks, and extracts the characteristics of the network topology (eigenvector centrality). The advantages of using this framework are: first, the number of variables used in this paper is large, and the dimensionality catastrophe can be overcome using the LASSO method. Second, this paper uses rolling windows to portray the time-varying characteristics of volatility spillovers in pan-financial market, which shortens the sample period, and utilizing the LASSO method reduces the sample size requirement. Third, the use of the DY spillover

index makes the results of the variance decomposition no longer dependent on the order of variables and also allows us to observe the interactive spillover effects among firms, identify the center of the systemic risk contagion network of China's pan-financial market, and the position of each firm in the network.

## LASSO-VAR model

The generalized prediction error variance decomposition results based on the VAR model can be used to measure the spillover effects among firms. Therefore, the paper constructs a VAR model to quantify the correlation between firms in the China's pan-financial market. Using the stock volatility of each company as an endogenous variable in the VAR model, an $N$-element $p$-order VAR model is defined, which can be expressed as:

$$Y_t = v + \sum_{i=1}^{p} \Phi_i Y_{t-i} + \varepsilon_t, \ t = 1, 2, \ldots, T \tag{1}$$

Where $Y_t$ is the stock volatility of the 112 listed companies included in pan-financial market in period $t$, $v$ is the 112×1 -dimensional intercept column vector, $\Phi_i$ denotes the 112×112-dimensional coefficient matrix, $\varepsilon_t$ is the 112×112-dimensional independently and identically distributed disturbance term, and $\Sigma$ is the covariance matrix. The number of parameters to be estimated is $112^2 \times p + 112$. When there are a large number of endogenous variables in the VAR model, the number of parameters will grow flatly with the number of variables, which means that the traditional VAR model will have insufficient degrees of freedom, i.e., the "curse of dimensionality" problem. In order to solve the problem of high-dimensional data dimensionality, the LASSO method is introduced to estimate the model parameters.

The basic idea of the LASSO method is to impose constraints on the sum of absolute values using regularization and to unify the parameter estimation and variable selection through the penalty term, which is realized synchronously so that the smaller coefficients are compressed to 0, thus obtaining a model with fewer degrees of freedom. Nicholson et al. [47] gave the LASSO-VAR model with different penalty forms, and this paper refers to their method and sets the LASSO-VAR model in the following form:

$$min \sum_{t=1}^{T} \parallel Y_t - v - \sum_{i=1}^{p} \Phi_i Y_{t-i} \parallel_F^2 + \lambda_i \parallel \Phi_i \parallel_1 \tag{2}$$

$$\parallel \Phi_i \parallel_1 = \sum_{j=1}^{N} |\Phi_{i,j}| \tag{3}$$

Where $\parallel Y_t - v - \sum_{i=1}^{p} \Phi_i Y_{t-i} \parallel_F$ is the Frobenius paradigm number of the matrix $Y_t - v - \sum_{i=1}^{p} \Phi_i Y_{t-i}$ and is the sum of the squares of the absolute values of all the elements of matrix $Y_t - v - \sum_{i=1}^{p} \Phi_i Y_{t-i}$. $\lambda_i \parallel \Phi_i \parallel_1$ denotes the penalty term, $\lambda_i$ is the penalty parameter controlling the size of the compression degree, and the rolling cross-validation method is used to determine the optimal $\lambda_i$, $\parallel \Phi_i \parallel_1$ denotes the 1-paradigm of the parameter to be estimated, which means that the result is robust when $\lambda_i$ is optimal.

Constructing listed firm volatility spillover networks for covariate-smooth $N$-dimensional variables $VAR(p)$ process (4), which is transformed into a moving average expression (5).

$$x_t = \sum_{i=1}^{p} \phi_i x_{t-i} + \varepsilon_t \tag{4}$$

$$x_t = \sum_{i=0}^{\infty} A_i \varepsilon_{t-i} \tag{5}$$

Where $\varepsilon \sim (0,\Sigma)$ is a perturbation vector with all components independently and identically distributed. $A_i$ is a matrix of $N \times N$ order coefficients obeying the following recursive process:

$$A_i = \phi_1 A_{i-1} + \phi_2 A_{i-2} + \cdots + \phi_p A_{i-p} \tag{6}$$

Where $A_0$ is a unit array of order $N$ and $A_i = 0$ when $i < 0$.

## DY spillover index

The contribution of the forecast error variance decomposition of the VAR model can be a helpful measure of spillover effects between variables. Given that the orthogonality assumption of the traditional Cholesky decomposition can make the prediction variance decomposition results very sensitive to the model order, the spillover indices constructed based on Cholesky decomposition are less robust. To address this issue, this paper uses the generalized variance decomposition proposed by Diebold and Yilmaz [42], which means that the dynamic interrelationships among variables are examined, i.e., the firm-to-firm contribution to the variance of the generalized variance decomposition of the overshooting H-step forecast. It expresses the extent to which variable changes are influenced by themselves or other variables in the system. The proportion of forecast error variance is the basis for constructing the variance decomposition spillover index in the following form:

$$d_{ij}^H = \frac{\sigma_{ii}^{-1} \sum_{h=0}^{H-1} \left( e_i A_h \sum e_j \right)^2}{\sum_{h=0}^{H-1} \left( e_i A_h \sum A_h e_i \right)} \tag{7}$$

Where $\Sigma$ denotes the covariance matrix of the disturbance vector $\varepsilon_t$, $H$ reflects the forecast period, and $h$ is the lag order of the perturbation term in Eq (7). $\sigma_{ii}$ is the standard deviation of $\varepsilon_t$, the $j$-th element of $e_j$ is 1, the remaining elements are 0, and $i,j = 1,\cdots\cdots,N, i \neq j$. Since the contributions of all endogenous variables of the generalized variance decomposition do not add up to 1, which means that $\sum_{j=1}^{N} d_{ij}^H \neq 1$, therefore, it is necessary to standardize each row of data:

$$\tilde{d}_{ij}^H = \frac{d_{ij}^H}{\sum_{j=1}^{N} d_{ij}^H} \tag{8}$$

Therefore, $\sum_{j=1}^{N} \tilde{d}_{ij}^H = 1$ and $\sum_{i,j=1}^{N} \tilde{d}_{ij}^H = N.\tilde{d}_{ij}^H$ denote the variance decomposition where the $j$-th endogenous variable is a shock and the $i$-th endogenous variable predicts the $H$-period. Then, the variance decomposition matrix $D_{ij}(h)$ of order $N \times N$ based on the composition of $\tilde{d}_{ij}^H$ can be used to capture the volatility risk spillover effect among different firms in pan-financial market. This means that the volatility spillover approach is used as a framework for risk contagion analysis.

$$D_{ij}(h) = \begin{pmatrix} d_{11} & \cdots & d_{1N} \\ \vdots & \ddots & \vdots \\ d_{N1} & \cdots & d_{NN} \end{pmatrix} \tag{9}$$

In the variance decomposition matrix $D_{ij}(h)(i \neq j)$, the off-diagonal elements represent the decomposition of the variance of the prediction error, reflecting the degree of risk spillover between firms $i$ and $j$. Thus, the sum of the rows represents the risk spillover of all other firms to it, indicating the risk tolerance of the firm. Thus, the sum of row $i$ in $D_{ij}(h)(i \neq j)$ represents the risk spillover to it from all other $j = 1, \cdots, N$ firms, denoting the risk tolerance of the firm. Further, from the perspective of a complex network, the nodes represent a particular company. The connecting edges between the nodes represent the risk spillover relationship between the companies. The variance contribution calculated by the variance decomposition is used as the weight of the volatility spillover network to construct the volatility spillover network for pan-financial market.

This means that the volatility spillover approach is used as a framework for risk contagion analysis.

$$C^H_{\cdot \leftarrow i} = \frac{\sum_{\substack{j=1 \\ i \neq j}}^{N} \tilde{d}^H_{ji} = 1}{\sum_{j=1}^{N} \tilde{d}^H_{ji} = 1} \tag{10}$$

Second, the volatility of each firm ($i$) receives a spillover effect caused from all other firms called the spillover effect received (FROM-Spillover), which can be expressed as:

$$C^H_{i \leftarrow \cdot} = \frac{\sum_{\substack{j=1 \\ i \neq j}}^{N} \tilde{d}^H_{ij} = 1}{\sum_{j=1}^{N} \tilde{d}^H_{ij} = 1} \tag{11}$$

Next, a thresholding approach is used to extract adequate information from the network obtained by the Pearson correlation coefficient. Considering the study's object and methodology and the network nodes' characteristics in the empirical section below, the paper chooses the 0.4 quartile of all spillover effects as the threshold value [48]. The prediction generalized variance decomposition matrix retains the edges with spillover effects that are more significant than the threshold value and deletes the edges below the threshold value. On this basis, each firm is regarded as a node in the volatility spillover network. The inter-firm risk spillover relationship is taken as the connecting edge of the complex network. The contribution of variance calculated by generalized prediction error variance decomposition is used as an adjacency matrix of the volatility spillover network to construct the net pairwise spillover network of China's pan-financial market.

## Systemic risk measurement based on complex network characterization indicators

LASSO-VAR-DY can not only effectively identify risky shocks within a complex network but also accurately measure the risk contribution of each node to the overall network, which is a better balance between macro-level and micro-level information, in order to comprehensively measure the systemic financial risk in the cross-sectional and time dimensions. Network centrality indicators can directly measure how vital financial institutions are in volatile spillover networks and give a better picture of how different individual financial institutions are [42,49]. Instead of focusing only on the more "local" nodes, Eigenvector centrality considers the reverse influence of other nodes from the perspective of the network as a whole and finds the nodes at the network's core. The metric not only measures the number of connected edges of a node but also takes the connected edge weight attribute into account, which means that eigenvector

centrality defines the influence of a node based on the influence ability of its neighboring nodes [50]. Therefore, the complex network eigenvector centrality approach is applied to measure the systemic risk contribution of each firm [51], expressed explicitly as:

$$S_{it} = \frac{1}{\lambda} \sum_{j=1}^{n} S_{jt} \rho_{ijt} \tag{12}$$

Where $S_{it}$ denotes the systematic risk contribution of firm $i$ in period $t$; $\rho_{ijt}$ denotes the matrix of correlation coefficients of shocks between firms at different time points. The equation is expressed as a matrix as $\lambda S = PS$, $\lambda$ is the eigenvalue corresponding to the eigenvectors, and the elements of the main diagonal of the matrix $P$ are all 0 as only the correlations between firms are taken into account and not their correlations.

## An empirical model of temperature difference affecting systemic risk in pan-financial market

**Benchmark regression model of temperature difference affecting systemic risk in pan-financial market.** Zhu and Ma [49] point out that eigenvector centrality can be an influential proxy variable for systemic risk. The paper uses this as a measure of systemic risk. It examines the impact of temperature difference on systemic risk in China's pan-financial market by combining temperature difference with listed firm-level and macro-level factors. The variables of firm characteristics referred to by Guo [52] and the variables of scale, leverage, and turnover ratio are selected. The scale of the firm, which means the market capitalization, can directly reflect the influence of the firm on the market; leverage is the ratio of total assets to owner's equity, which measures a firm's own financial risk; and turnover ratio is the frequency of stock turnover and trading, which measures stock liquidity. Macroeconomic variables are referred to by Adrian and Brunnermeier [53] and selected variables such as the China real estate climate index, consumer price index, short-term liquidity spread, term structure of interest rate, and credit spread, which means that the difference between the yield to maturity of 10-year treasury bonds and 3-month treasury bonds reflects the expectations and confidence of market participants in the future; and credit spread, which means that the difference between the yield to maturity of 10-year corporate bonds (AAA) and the yield on 10-year treasury bonds reflects the risk of credit default of real enterprises.

Establishing a benchmark regression, in addition to the firms' characteristics considered in the model, managerial ability differences, the degree of appetite for executive risk, and other variables that are not suitable for observation are also essential factors affecting systemic risk, and these can be explained to some extent by controlling for firm fixed effects. In addition, there may be various macro shocks in different years, such as financial crises, policy adjustments, economic operating cycles, public health emergencies, major natural disasters, etc., and the outbreak of these uncertain events makes the stock volatility of the firms affected. The incorporation of time-fixed effects overcomes these shocks to a certain extent, and this approach also refers to the research of George et al. [54]. Therefore, the following two-way fixed effects panel regression model is developed in this paper:

$$Sysrisk_{i,t} = \alpha_0 + \beta Temp_{i,t} + \sum_{m=1}^{3} \gamma_m Firm_{i,t,m} + \sum_{n=1}^{4} \delta_n Macro_{t,n} + \theta_i + \lambda_t + \varepsilon_{i,t} \tag{13}$$

Where $Sysrisk_{i,t}$ is the systematic risk of firm $i$ at time $t$; $Temp_{i,t}$ is the temperature difference at time $t$ for the prefecture-level city where firm $i$ is located; $Firm_{i,t,m}$ is a firm-level characterization variable, and $m = 1,2,3$ denotes the scale, leverage, and turnover of firm $i$ at time $t$,

respectively; $Macro_{t,n}$ is a macroeconomic level variable, and $n = 1,2,3,4$ denotes the China real estate climate index, consumer price index, short-term liquidity spread, and the term structure of interest rate, respectively; $\theta_i$ are individual effects; $\lambda_t$ are time effects; and $\varepsilon_{i,t}$ are randomized perturbation terms.

## Subsectoral regression model of temperature difference affecting systemic risk in pan-financial market

Based on the benchmark model and pan-financial market constructed in the paper (including banking, securities, insurance, real estate, traditional energy, and new energy sectors), group regressions are performed to analyze the differences in the impact of temperature difference on the systemic risk of firms within each sector and to combine the analysis with firm characteristics and macroeconomic characteristics.

$$Sysrisk_{k,i,t} = \alpha_{k,0} + \varphi_k Temp_{k,i,t} + \sum_{m=1}^{3} \gamma_{k,m} Firm_{k,i,t,m} + \sum_{n=1}^{4} \delta_{k,n} Macro_{k,t,n} + \theta_{k,i} + \lambda_{k,t} + \varepsilon_{k,i,t} \quad (14)$$

Where $k = 1,2,3,4,5,6$ represent the banking, securities, insurance, real estate, traditional energy, and new energy sectors, respectively; $Sysrisk_{k,i,t}$ is the systematic risk of firm $i$ in sector $k$ at time $t$; $Temp_{k,i,t}$ is the temperature difference at time $t$ in the prefecture where firm $i$ in sector $k$ is located; $Firm_{k,i,t,m}$ is a firm-level characterization variable that includes the scale, leverage, and turnover of firm $i$ in sector $k$ at time $t$; $Macro_{k,t,n}$ are macroeconomic level variables, including the China real estate climate index, the consumer price index, the short-term liquidity spread and the term structure of interest rate; $\theta_{k,i}$ are individual effects; $\lambda_{k,t}$ are time effects; and $\varepsilon_{k,i,t}$ are randomized perturbation terms.

To summarize the above research and design department flowchart, as shown in Fig 1.

## Measurement and analysis of temperature difference and systemic risk in pan-financial market

### Selection of the China's pan-financial market data

In the context of "carbon neutrality," sectoral organizations are not an "island" during temperature change, and the exposure of traditional financial institutions and the real sector to climate risk is escalating. In the context of carbon neutrality, the most critical driver for accelerating the energy mix transition is no longer the economic efficiency of new energy sources but rather a response to temperature change. The policy documents promulgated by the state on the transformation of the high-carbon industry have affected physical enterprises. The tightened environmental protection policies have increased the operating costs of the high-carbon industry. The enterprises are facing various risks, such as the depreciation of assets, decline in profitability, and increase in disputes, which have forced the firm to adjust its business strategies, change its production mode, and comply with the relevant regulations on energy conservation and emission reduction. The systematic risks for the financial institutions that have business dealings with this type of industry may be increased. Given this, the paper selects the banking, securities, and insurance sectors of the traditional financial system and the real estate, traditional energy, and new energy sectors of the susceptible industries, which together comprise China's pan-financial market, as a means of investigating the impact of temperature difference on the systemic risk of China's pan-financial market.

Due to the frequency of extreme weather events in recent years, temperature changes are a mainstay affecting the stability of the entire financial system. They can also have an impact on macroprudential regulation. Therefore, the empirical study of the paper focuses on the banking,

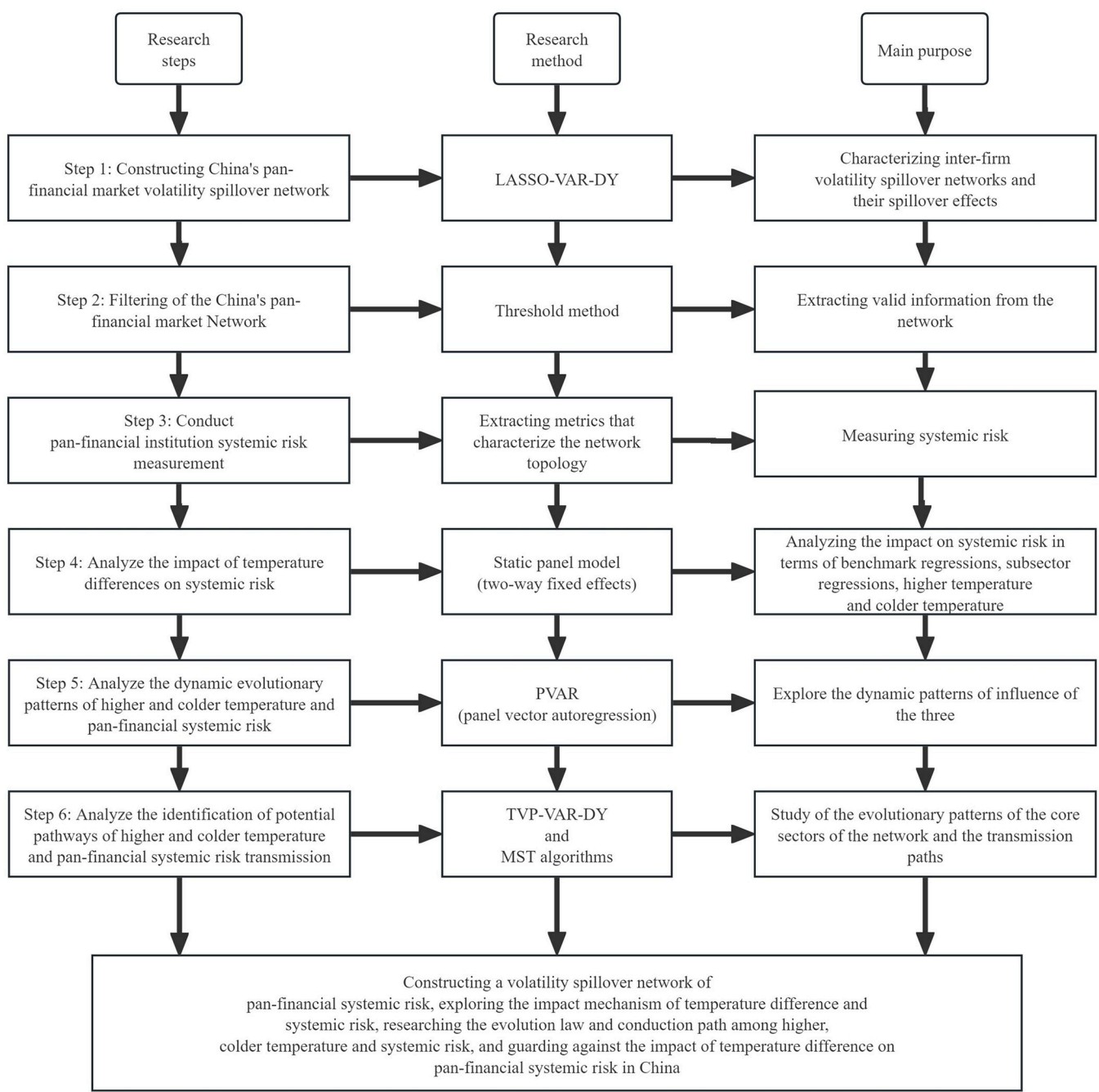

**Fig 1. Research methods.**

securities, insurance, real estate, traditional energy, and new energy sectors. First, a volatility spillover network of China's listed pan-financial market is constructed, from which the eigenvector centrality is extracted as the systematic risk of each firm. Second, the impact of temperature difference on the overall systemic risk of China's pan-financial market is analyzed by considering macro-level factors and firm-level factors. Third, the impact of temperature difference on the systemic risk of China's pan-financial market is further analyzed in a comparative manner by sector.

According to the Wind Sector Index, as of June 30, 2022, the constituents include forty-three banks, twenty-one securities companies, six insurance companies, one hundred and twenty real estate companies, fifty-eight new energy companies, and eighty-one traditional energy companies, among other listed companies. Due to the late listing of some companies, the data needs to be completed, and companies listed after January 4, 2010, are excluded. The sample is determined based on including the Corona Virus Disease 2019 (COVID-19) shocks and the requirement of being continuously listed and traded during the sample period. A total of one hundred and twelve listed companies were selected from January 4, 2012, to June 30, 2022, including fifteen banks, eleven securities companies, five insurance companies, twenty-nine real estate companies, twenty new energy companies, and thirty-two traditional energy companies, whose total assets accounted for more than 75% of the whole industry, so the sample is representative. The detailed list is shown in Table 1.

**Table 1. List of the China's pan-financial market.**

| Sector | Banking | | Real Estate | | New Energy | | Traditional Energy | |
|---|---|---|---|---|---|---|---|---|
| Number | Code | Abbreviation | Code | Abbreviation | Code | Abbreviation | Code | Abbreviation |
| 1 | 601169 | BOB | 002244 | BJJT | 002610 | AKKJ | 601011 | BTL |
| 2 | 601398 | ICBC | 600649 | CTKG | 000690 | BXNY | 002221 | DHNY |
| 3 | 601818 | CCPB | 600185 | GLDC | 600875 | DFDQ | 000096 | GJNY |
| 4 | 600015 | HXB | 000886 | HNGS | 300118 | DFRS | 000159 | GJSY |
| 5 | 601939 | CCB | 000861 | HYGF | 600151 | HTJD | 300084 | HMKJ |
| 6 | 601328 | BCM | 600325 | HFGF | 002056 | HDDC | 600583 | HYGC |
| 7 | 600016 | CMBC | 600503 | HLJZ | 000875 | JDGF | 601101 | HHNY |
| 8 | 601009 | NJCB | 000036 | HLKG | 002202 | JFKJ | 300157 | HTAP |
| 9 | 002142 | NBCB | 600340 | HXXF | 601908 | JYT | 600971 | HYMD |
| 10 | 601288 | ADBC | 600383 | JDJT | 601727 | SHDQ | 002554 | HBP |
| 11 | 600000 | SPDB | 000656 | JKGF | 002218 | TRXN | 000937 | JZNY |
| 12 | 601166 | CIB | 000402 | JRJ | 600089 | TBDG | 002353 | JRGF |
| 13 | 600036 | CMB | 600663 | LJZ | 002531 | TSFN | 000552 | JYMD |
| 14 | 601988 | BOC | 600064 | NJGK | 600438 | TWGF | 600997 | TLGF |
| 15 | 601998 | CITIC | 600790 | QFC | 300185 | TYZG | 600123 | LHKC |
| 16 | | | 002146 | RSFZ | 002623 | YMD | 601699 | LAHN |
| 17 | | | 000006 | SZYA | 300274 | YGDY | 000723 | MJNY |
| 18 | | | 600376 | SKGF | 600537 | YJGD | 600395 | PJGF |
| 19 | | | 000631 | SFHY | 601877 | ZTDQ | 601666 | HMGF |
| 20 | Securities | | 000718 | SNHQ | 002080 | ZCKJ | 300191 | QNHX |
| 21 | Code | Abbreviation | 600846 | TJKJ | | | 600740 | SXJH |
| 22 | 300059 | DFCF | 000002 | WKA | | | 002267 | STRQ |
| 23 | 601555 | DWZQ | 600641 | WYQY | | | 600508 | SHNY |
| 24 | 601788 | GDZQ | 600773 | XZCT | | | 002278 | SKGF |
| 25 | 000776 | GFZQ | 600208 | XHZB | | | 000554 | TSSY |
| 26 | 600109 | GJZQ | 600657 | XDDC | Insurance | | 300164 | TYSY |
| 27 | 000728 | GYZQ | 000671 | YGC | Code | Abbreviation | 600792 | YMNY |
| 28 | 600837 | HTZQ | 600895 | ZJGK | 000627 | TMJT | 601088 | SHNY |
| 29 | 601688 | HTZQ | 000961 | ZNJS | 601336 | NCI | 600028 | ZGSH |
| 30 | 601377 | XYZQ | | | 601318 | PAIC | 601857 | ZGSY |
| 31 | 600999 | ZSZQ | | | 601628 | GPIC | 601808 | ZHYF |
| 32 | 600030 | ZXZQ | | | 601601 | CPIC | 601898 | ZMNY |

## Selection and measurement of temperature difference

The level of long-term temperature change is often used to refer to climate change [47], and according to Kahn et al. [55], we used the difference in deviation from the historical mean temperature as a proxy variable for temperature difference. The European Center for Medium-Range Weather Forecasts (ECMWF) published the ERA5-Land, which provided the temperature data for each region of China. Considering the extensive latitude and longitude spans in China and the apparent temperature difference across regions, we first counted the monthly averages of the mean surface air temperatures in the regions where the 112 listed companies are located and eliminated the trend by subtracting the historical standard from the monthly averages, which reflects the monthly temperatures that deviate from the historical standard, which is calculated using the H = 120 / 80 / 40 monthly moving average, resulting in a variable temperature difference variable *Temp*.

Temperature deviations from historical averages primarily represent de-trended fluctuations in temperatures. However, short-term temperatures may be above or below their historical norms, and whether higher or colder temperature has the same impact on systemic risk in pan-financial market. To answer this question, we decompose temperature deviations from historical averages into two components: *Temp_plus* = *max*(*Temp_plus*, 0) means retaining temperature more higher than the historical average and setting the rest of the observations to zero; *Temp_minus* = *min*(*Temp_plus*, 0) means retaining temperature colder than the historical average and setting the rest of the observations to zero. In this way *Temp_plus* = *max*(*Temp_plus*, 0) and *Temp_plus* = *max*(*Temp_plus*, 0)s represent periods hotter and colder than historical temperature, respectively, to empirically investigate the asymmetric effects of higher and colder temperature on the impact of systemic risk.

## Selection and characterization of systemic risk in the China's pan-financial market

The paper measures the systemic risk of individual firms based on economic and financial correlation networks. For the measure of volatility, it is mainly based on the actual volatility of the stock [43,56], defined as:

$$\sigma_{it}^2 = 0.361(log(p_{it}^{high}) - (p_{it}^{low}))^2, \ i = 1, \ldots, N \tag{15}$$

where $p_{it}^{high}$ and $p_{it}^{low}$ are firm i highest and lowest stock prices on day t, respectively, and N is the total number of firms. The stock volatility data of each company selected in this paper is daily frequency, and the sample interval is from January 1, 2012, to June 1, 2022. Meanwhile, this paper adopts the rolling window method in constructing the LASSO-VAR-DY model, choosing a window period of 200 periods (1 year), a maximum lag period of 4 periods, and a variance decomposition forecast period of 10 periods. The rolling window method will lose 200 sample periods, and the final calculated volatility spillover effect takes the value from January 1, 2013, to June 30, 2022.

The following conclusions can be drawn from the trend of changes in systemic risk in pan-financial market from 2013 to 2022 (Fig 2): First, with the arrival of the bull market in 2013, the degree of inter-firm correlation became closer. Systemic risk gradually increased, spreading risk to pan-financial market through balance sheets and other channels until the Chinese stock market bubble burst in June 2015, when inter-firm correlation began to weaken, and systemic risk declined. Second, liquidity risks triggered panic when China's RMB common stock market melted in 2016. At the same time, housing prices in some cities skyrocketed, which led the real estate market into a high-pressure situation. Risks continued to spread across market,

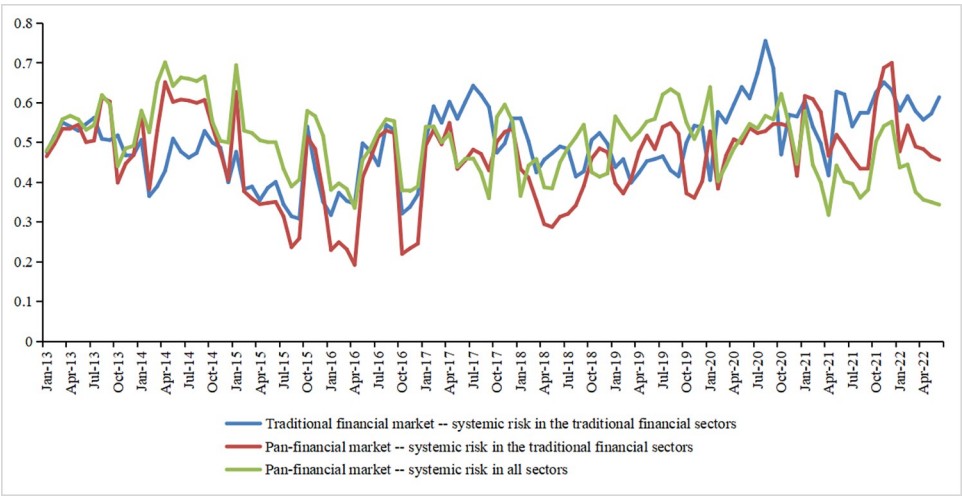

**Fig 2. Trends in systemic risk in traditional finance versus pan-financial market from 2013 to 2022.**

and systemic risks rose again. Third, to promote the financial market's healthy development, China's financial industry issued more than 20 crucial regulatory documents in 2017, which was referred to by the financial sector as the strictest year of regulation in history. With the gradual implementation of regulatory policies and the intensification of deleveraging, the systemic risk of China's pan-financial market has gradually receded. Fourth, the China-US trade war continued to intensify in 2018. Cooperation in energy fields such as power equipment, oil and gas, photovoltaic products, and new energy vehicles was affected. It subsequently exacerbated risks in energy and financial market, further increasing systemic risks in China. Fifth, both the financial and real economies in China have been hit hard by the impact of the Corona Virus Disease 2019 (COVID-19), with a pause in the economy in early 2020, followed by a severe setback in the second half of the year, increased risk of contagion among firms in pan-financial market, and heightened systemic risk.

Fig 2 compares the mean values of systemic risk with and without the banking, insurance, and securities sectors in pan-financial market. This shows that the overall monthly mean value of systemic risk in the banking, insurance, and securities sectors decreases after including pan-financial market. Specifically, the monthly mean value of systemic risk in the traditional financial sector is 0.499, and the monthly mean value of systemic risk after including pan-financial market is 0.462. Since the paper's measure of systemic risk is the eigenvector centrality of the network topology, it suggests that pan-financial market, which includes the real estate, traditional energy, and new energy sectors, can reduce the connectivity of the traditional financial sector, which in turn reduces the systemic risk of the traditional financial sectors.

## Descriptive analysis of variables

Table 2 shows the descriptive statistics for each variable, including the mean, standard deviation, maximum, minimum, and unit root test results for the systematic risk, temperature difference, higher temperature, and colder temperature, the micro-control variables of scale, turnover rate, leverage, and the macro-control variables of the China real estate climate index, consumer price index, term structure of interest rate, short-term liquidity spread, and credit spread.

The standard deviation of systematic risk is 0.166, indicating some variation among firms. Temperature difference, rise, and fall remain stable in their historical criteria of 40, 80, and 120

**Table 2. Descriptive statistics of variables.**

| Variable | Abbreviations | Observations | Mean | Std. Dev | Minimum | Maximum | Unit Root Test |
|---|---|---|---|---|---|---|---|
| Systemic Risks | Sysrisk | 12,768 | 0.4994 | 0.1655 | 0.0965 | 1.0000 | stable |
| Temperature Difference(40) | Temp40 | 12,768 | 0.0029 | 0.4547 | -1.2904 | 1.0474 | stable |
| Temperature Difference(80) | Temp80 | 12,768 | 0.0032 | 0.2413 | -0.6733 | 0.5627 | stable |
| Temperature Difference(120) | Temp120 | 12,768 | 0.0024 | 0.1568 | -0.4432 | 0.3708 | stable |
| Higher Temperature(40) | Temp_plus40 | 12,768 | 0.1942 | 0.2450 | 0.0000 | 1.0474 | stable |
| Colder Temperature(40) | Temp_minus40 | 12,768 | 0.1914 | 0.2690 | 0.0000 | 1.2904 | stable |
| Higher Temperature(80) | Temp_plus80 | 12,768 | 0.1039 | 0.1316 | 0.0000 | 0.5627 | stable |
| Colder Temperature(80) | Temp_minus80 | 12,768 | 0.1008 | 0.1414 | 0.0000 | 0.6733 | stable |
| Higher Temperature(120) | Temp_plus120 | 12,768 | 0.0677 | 0.0856 | 0.0000 | 0.3708 | stable |
| Colder Temperature(120) | Temp_minus120 | 12,768 | 0.0653 | 0.0918 | 0.0000 | 0.4432 | stable |
| Scale | Scale | 12,768 | 136.1172 | 314.9212 | 1.3710 | 2,502.15 | stable |
| Turnover Rate | Turn | 12,768 | 1.1816 | 1.4504 | 0.0000 | 22.6622 | stable |
| Leverage | Lever | 12,768 | 0.3775 | 0.2255 | 0.0462 | 0.9803 | stable |
| China Real Estate Climate Index | CRECI | 12,320 | 98.9265 | 2.9297 | 92.4300 | 102.0300 | stable |
| Consumer Price Index | CPI | 12,768 | 101.9833 | 0.9409 | 99.5000 | 105.4000 | stable |
| Term Structure Of Interest Rate | Rate | 12,768 | 0.7508 | 0.3570 | -0.1381 | 1.7474 | stable |
| Short-term Liquidity Spread | Short | 12,768 | 0.8527 | 0.4428 | -0.0436 | 1.9664 | stable |
| Credit Spread | Credit | 12,768 | 1.2010 | 0.3072 | 0.2812 | 1.8957 | stable |

periods samples, indicating that the sample selection is reasonable. The firms' scales have the most significant standard deviation, indicating that the selected sample has comprehensive and representative coverage. The macro-control variables leverage ratio, China real estate climate index, consumer price index, term structure of interest rate, short-term liquidity spread, and credit spread have more minimum standard deviations, meaning the data are stabler. Since the period of the panel data is much larger than its cross-sectional span, the paper performs unit root tests on all the variables in the equation, resulting in stable variables that can be regressed in the baseline model.

## Empirical analysis of the mechanisms influencing systemic risk in pan-financial market

### Benchmark estimates of the mechanisms influencing temperature difference and systemic risk in pan-financial market

Table 3 shows the results of the baseline regression, which has columns (1), (2), and (3) as two-way fixed-effects regressions with temperature difference as the core independent variable based on the historical standardized periods of 40, 80, and 120, respectively. It has columns (4), (5), and (6) as two-way fixed-effects regressions with higher and colder temperature as the core independent variables based on the historical standardized periods of 40, 80, and 120, respectively.

Mechanism of temperature difference on systemic risk: Some scholars have demonstrated that higher temperature causes more damage in geographic regions with hotter climates [57]. Dell et al. [58] found that higher temperature leads to a significant decline in economic growth in developing countries, which can cause more damage in geographic regions with hotter climates. Kahn et al. [55] used an extended dataset to show that the long-term macroeconomic impacts of weather anomalies are uneven across countries and that the response of economic growth to temperature is nonlinear. Meanwhile, higher temperature reduces labor

**Table 3. Benchmark regression results.**

| | Dependent variable: systemic risk of pan-financial market | | | | | |
|---|---|---|---|---|---|---|
| | **(1)** | **(2)** | **(3)** | **(4)** | **(5)** | **(6)** |
| Temp40 | 0.0240*** | | | | | |
| | (2.7562) | | | | | |
| Temp80 | | 0.0444*** | | | | |
| | | (2.7454) | | | | |
| Temp120 | | | 0.0738*** | | | |
| | | | (3.0044) | | | |
| Temp_plus40 | | | | 0.0384** | | |
| | | | | (2.4049) | | |
| Temp_minus40 | | | | -0.0096* | | |
| | | | | (-0.7749) | | |
| Temp_plus80 | | | | | 0.0551** | |
| | | | | | (1.8473) | |
| Temp_minus80 | | | | | -0.0338* | |
| | | | | | (-1.4560) | |
| Temp_plus120 | | | | | | 0.0750* |
| | | | | | | (1.6656) |
| Temp_minus120 | | | | | | -0.0727** |
| | | | | | | (-2.0479) |
| Scale | -0.0000*** | -0.0000*** | -0.0000*** | -0.0000*** | -0.0000*** | -0.0000*** |
| | (-0.6499) | (-0.6514) | (-0.6546) | (-0.6486) | (-0.6508) | (-0.6545) |
| Turn | -0.0151*** | -0.0151*** | -0.0151*** | -0.0151*** | -0.0151*** | -0.0151*** |
| | (-6.1340) | (-6.1324) | (-6.1334) | (-6.1297) | (-6.1268) | (-6.1264) |
| Lever | 0.0817* | 0.0814* | 0.0813* | 0.0817* | 0.0815* | 0.0813* |
| | (1.8711) | (1.8652) | (1.8602) | (1.8726) | (1.8656) | (1.8601) |
| CRECI | -0.0079*** | -0.0080*** | -0.0080*** | -0.0079*** | -0.0080*** | -0.0080*** |
| | (-7.9004) | (-7.9412) | (-7.9648) | (-7.8976) | (-7.9321) | (-7.9556) |
| CPI | 0.0073*** | 0.0073*** | 0.0073*** | 0.0073*** | 0.0073*** | 0.0073*** |
| | (2.6877) | (2.7021) | (2.6909) | (2.7062) | (2.7127) | (2.6921) |
| Rate | 0.0287*** | 0.0287*** | 0.0286*** | 0.0287*** | 0.0287*** | 0.0286*** |
| | (4.8614) | (4.8615) | (4.8430) | (4.8645) | (4.8566) | (4.8400) |
| Short | 0.0311*** | 0.0311*** | 0.0313*** | 0.0310*** | 0.0311*** | 0.0313*** |
| | (3.3599) | (3.3611) | (3.3816) | (3.3388) | (3.3502) | (3.3794) |
| Credit | 0.0841*** | 0.0845*** | 0.0844*** | 0.0839*** | 0.0844*** | 0.0844*** |
| | (6.3967) | (6.4187) | (6.4125) | (6.3795) | (6.4133) | (6.4074) |
| Constant | 0.4336** | 0.4325** | 0.4387** | 0.4218** | 0.4278** | 0.4384** |
| | (2.5133) | (2.5071) | (2.5418) | (2.4394) | (2.4731) | (2.5335) |
| Individual Effect | YES | YES | YES | YES | YES | YES |
| Time Effect | YES | YES | YES | YES | YES | YES |
| Observations | 12,320 | 12,320 | 12,320 | 12,320 | 12,320 | 12,320 |
| R-squared | 0.0827 | 0.0827 | 0.0828 | 0.0828 | 0.0827 | 0.0828 |
| Number Of Firms | 112 | 112 | 112 | 112 | 112 | 112 |

Note

***, ** and *denote significance at 1%, 5% and 10% levels, respectively; t-statistics in parentheses.

productivity [59] and macroeconomic growth and negatively affects assets [60], and these scholars' studies are consistent with the empirical results of this paper. This paper shows that temperature difference positively relates to systemic risk in pan-financial market, which means systemic risk is exacerbated as the temperature difference increases. The coefficients of temperature difference based on the historical norms of 40, 80, and 120 periods gradually increase, suggesting that the longer the period span, the more significant the impact of temperature change on systemic risk. Specifically, higher temperature will exacerbate the impact on systemic risk, suggesting that hot weather will lead to losses in production and operations, resulting in credit delinquency and exacerbating systemic risk in pan-financial market; colder temperature will mitigate the impact on systemic risk, suggesting that colder temperature will mitigate the phenomena of glacier melting and sea level rise, which will effectively curb the systemic risk in pan-financial market. Taken together, higher and colder temperature has an asymmetric effect on the systemic risk of pan-financial market.

For the control variables at the individual firm level, scale, turnover rate, and leverage can reflect the firm's operating conditions and activity level. The significantly negative effect of firm scale may be that larger firms have a tremendous advantage in deferring systemic risk because of advanced risk management, extensive management experience, and strict shareholder oversight rules. The turnover rate has a negative impact. This indicator reflects the firm's trading activity. A higher turnover rate indicates that the firm's operators are more likely to detect market turbulence and thus adjust their strategies to reduce systemic risk. There is a significant relationship between a firm's leverage and its systemic risk, and the positive results of this regression indicate that an increase in leverage is conducive to expanding earnings. However, too much leverage exacerbates one's financial risk, leading to high earnings while amplifying the systemic risk of pan-financial market.

For the macro-control variables, the China real estate climate index has a significantly negative impact on systemic risk, reflecting the operating conditions of the macroeconomy, and an increase in this index would favor the balance sheet and reduce the incidence of default risk. The consumer price index, which reflects changes in purchasing power, has a positive impact, suggesting that higher levels of consumption may lead to price increases and even inflation, ultimately resulting in the devaluation of currencies, increased risk contagion among sectoral firms, and an exacerbation of systemic risks in pan-financial market. The term structure of interest rate measures the future expectations and confidence of market participants. An increase in the term structure of interest rate indicates positive economic expectations. Investment behavior will increase the degree of inter-firm correlation, and over-investment may lead to systemic risk. Short-term liquidity spread positively impacts and measures liquidity risk in financial market, where over-reliance on short-term funding can lead to a sharp increase in liquidity risk across the system, triggering systemic risk. Credit spread reflects a firm's default probability and positively correlates with systemic risk.

## Sub-sectoral estimation results of the mechanisms affecting temperature difference and systemic risk in pan-financial market

Table 4 shows the results of the sub-sample regressions for the banking, securities, insurance, real estate, new energy, and traditional energy sectors, respectively, using the temperature difference as the core independent variable based on the historical norms of 40, 80, and 120 periods.

Some scholars have shown that extreme temperature difference can affect organizational structures, operations, supply chains, transportation needs, and employee safety. Some insurance companies also bear significant losses due to temperature difference, thus affecting the quality of the institutions' assets. In addition, policy goals for the transition to a low-carbon

**Table 4. Sub-sector regression results with temperature difference as the core independent variable.**

| | Dependent variable: systemic risk of pan-financial market | | | | | |
|---|---|---|---|---|---|---|
| | **(1)** | **(2)** | **(3)** | **(4)** | **(5)** | **(6)** |
| | **Banking** | **Securities** | **Insurance** | **Real Estate** | **New Energy** | **Traditional Energy** |
| Temp40 | 0.0248 | 0.0782* | 0.0144 | -0.002 | 0.0356** | 0.0450** |
| | (0.9481) | (1.7471) | (0.4360) | (-0.1237) | (1.9764) | (2.1342) |
| Constant | 5.2609*** | 1.2848** | 2.0334*** | 0.8597*** | -1.1369*** | -1.9818*** |
| | (10.8593) | (2.3316) | (3.2304) | (2.7470) | (-3.0999) | (-5.7626) |
| R-squared | 0.2467 | 0.1124 | 0.1856 | 0.0787 | 0.198 | 0.1581 |
| Temp80 | 0.0485 | 0.1402 | 0.0313 | -0.0039 | 0.0602* | 0.0766* |
| | (0.9727) | (1.6277) | (0.4964) | (-0.1273) | (1.7528) | (1.9024) |
| Constant | 5.2607*** | 1.2805** | 2.0345*** | 0.8595*** | -1.1413*** | -1.9896*** |
| | (10.8611) | (2.3220) | (3.2343) | (2.7462) | (-3.1107) | (-5.7838) |
| R-squared | 0.2467 | 0.1121 | 0.1857 | 0.0787 | 0.1977 | 0.1578 |
| Temp120 | 0.0777 | 0.2343* | 0.0649 | 0.0015 | 0.0930* | 0.1228** |
| | (1.0182) | (1.7868) | (0.6702) | (0.0321) | (1.7656) | (1.9930) |
| Constant | 5.2684*** | 1.2992** | 2.0468*** | 0.8631*** | -1.1379*** | -1.9790*** |
| | (10.8663) | (2.3541) | (3.2523) | (2.7568) | (-3.1006) | (-5.7462) |
| R-squared | 0.2468 | 0.1125 | 0.186 | 0.0787 | 0.1978 | 0.1579 |
| Control Variables | YES | YES | YES | YES | YES | YES |
| Individual Effect | YES | YES | YES | YES | YES | YES |
| Time Effect | YES | YES | YES | YES | YES | YES |
| Observations | 1650 | 1210 | 550 | 3190 | 2200 | 3520 |
| Number Of Firms | 15 | 11 | 5 | 29 | 20 | 32 |

Note

\***, \*\* and \*denote significance at 1%, 5% and 10% levels, respectively; t-statistics in parentheses.

economy will be more stringent due to drastic climate change, and a large number of carbon-sensitive firms will face a shrinking of their assets, which in turn affects the uncertainty that causes commodity prices and triggers stock market volatility [61]. At the industry level, temperature often affects the profitability of climate-sensitive industries such as agriculture, energy, tourism, and manufacturing [62]. The effect of temperature difference is significant in the securities and energy sectors, and the extent to which temperature difference positively affects systemic risk increases with the historical criterion, which means that the longer the period span of the temperature change, consistent with the total sample results, suggests that the securities sector, the new energy sector, and the traditional energy sector are the sectors that are highly susceptible to temperature change. With global warming, production decisions in the traditional energy sector are highly susceptible to hot weather. Implementing carbon neutral and peak carbon policies to reduce carbon emissions in high-carbon emitting sectors will trigger systemic risks in the traditional energy sector. Global warming means that research, development, and investment in new energy sources will further accelerate in the new energy sector. The uncertainty of new energy development will exacerbate systemic risk. In contrast, a series of systemic disasters, such as labor disruptions, power supply shortages, mountain fire crises, and the security of food supplies, will be reflected in the stock market as the higher temperature, thereby triggering systemic risk in the securities sector.

Table 5 shows the results of the sub-sample regressions for the banking, securities, insurance, real estate, new energy, and traditional energy sectors, using the historical standard 40, 80, and 120 periods of higher and colder temperature as the core independent variables.

**Table 5. Sub-sectoral regression results with higher and colder temperature as core independent variables.**

| | Dependent variable: systemic risk of pan-financial market | | | | | |
|---|---|---|---|---|---|---|
| | **(1)** | **(2)** | **(3)** | **(4)** | **(5)** | **(6)** |
| | **Banking** | **Securities** | **Insurance** | **Real Estate** | **New Energy** | **Traditional Energy** |
| Temp_plus40 | 0.0983* | -0.0109 | 0.0117 | 0.0231 | 0.0249 | 0.0570 |
| | (1.9474) | (-0.1287) | (0.2634) | (0.7208) | (0.9488) | (1.3797) |
| Temp_minus40 | 0.0526 | -0.1734*** | -0.0173 | 0.0261 | -0.0456* | -0.0323 |
| | (1.3191) | (-4.1241) | (-0.3258) | (1.3624) | (-1.8356) | (-1.3262) |
| Constant | 5.2007*** | 1.3613** | 2.0359*** | 0.8413*** | -1.1270*** | -1.9929*** |
| | (10.7181) | (2.4619) | (3.2213) | (2.6829) | (-3.0635) | (-5.7745) |
| R-squared | 0.2483 | 0.1142 | 0.1856 | 0.0789 | 0.1981 | 0.1581 |
| Temp_plus80 | 0.1954* | -0.0717 | 0.0052 | 0.0115 | 0.0078 | 0.0880 |
| | (2.0802) | (-0.4658) | (0.0561) | (0.1824) | (0.1882) | (1.1345) |
| Temp_minus80 | 0.1049 | -0.3604*** | -0.0589 | 0.0188 | -0.1090** | -0.0646 |
| | (1.3852) | (-4.2111) | (-0.6362) | (0.5192) | (-2.5586) | (-1.3864) |
| Constant | 5.1955*** | 1.3730** | 2.0472*** | 0.8534*** | -1.1154*** | -1.9955*** |
| | (10.7081) | (2.4820) | (3.2415) | (2.7204) | (-3.0308) | (-5.7772) |
| R-squared | 0.2484 | 0.1147 | 0.1858 | 0.0787 | 0.1980 | 0.1578 |
| Temp_plus120 | 0.2293* | -0.1521 | -0.0103 | 0.0099 | 0.0183 | 0.1470 |
| | (1.6912) | (-0.6510) | (-0.0699) | (0.1037) | (0.2931) | (1.2043) |
| Temp_minus120 | 0.0804 | -0.6314*** | -0.1444 | 0.0065 | -0.1618** | -0.0972 |
| | (0.7635) | (-4.8940) | (-1.0063) | (0.1163) | (-2.3379) | (-1.3243) |
| Constant | 5.2255*** | 1.4044** | 2.0698*** | 0.8610*** | -1.1135*** | -1.9870*** |
| | (10.7546) | (2.5396) | (3.2771) | (2.7438) | (-3.0244) | (-5.7470) |
| R-squared | 0.2476 | 0.1162 | 0.1864 | 0.0787 | 0.1980 | 0.1579 |
| Control Variables | YES | YES | YES | YES | YES | YES |
| Individual Effect | YES | YES | YES | YES | YES | YES |
| Time Effect | YES | YES | YES | YES | YES | YES |
| Observations | 1,650 | 1,210 | 550 | 3,190 | 2,200 | 3,520 |
| Number Of Firms | 15 | 11 | 5 | 29 | 20 | 32 |

Note

\*\*\*, \*\* and \*denote significance at 1%, 5% and 10% levels, respectively; t-statistics in parentheses.

In the banking sector, higher temperature makes systemic risk worse by a significant amount. This aligns with the total sample results, which suggest that the banking sector is most affected by higher temperature. This means that as temperature rises, the operations of the sectors are affected, which in turn sends systemic risk to the banking sector, which in turn causes systemic risk in the banking sector, and to a greater extent, as the length of the historical record increases. The temperature difference mainly affects the securities and new energy sectors. This is in line with the results in Table 4, which show that the effect of the temperature difference on these two sectors is mainly reflected in the inhibitory effect of the colder temperature. This means that the securities and new energy sectors are susceptible to the effect of the colder temperature. So, the colder temperature will help the development of the new energy sector.

The subsector regression results show that the coefficient estimates for most of the sectoral variables are consistent with the full-sample significance tests. The panel regression coefficients based on the micro-firm-level variables and the macroeconomic variables are generally consistent in magnitude, which suggests that the factors affecting systemic risk in pan-financial market are both the firms themselves as well as the industries in which they are embedded and the

macro-environment, such as the financial ecosystem. Although many reform measures have been implemented in China's pan-financial market, the current financial environment in China still needs to meet the needs of the healthy development of pan-financial market in order to serve the real economy better. Many aspects need to be improved, which means that they are related to the debt of the local government, the structure of the driving force of economic growth, the construction of the rule of law in finance, the fostering of a culture of social integrity, and other basic environments that are related to the survival of the main body of the financial sector. Therefore, the paper argues that the fundamentals of reducing systemic risk in pan-financial market have to be approached from a variety of perspectives, both in terms of guarding against shocks due to temperature difference and controlling leverage ratios at the micro level, as well as the overall level of economic performance at the macro level, the structure of interest rate, liquidity, credit spread, and other aspects.

## Robustness tests of the mechanism influencing temperature difference and systemic risk in pan-financial market

In order to test the robustness of the empirical results, the thesis uses six methods to test the robustness of the model. Firstly, removing macro control variables. Second, in the regression analysis, the variables are deflated by 5% to consider the presence of extreme values in the data. Third, considering the chance of sample time selection, the regression analysis is chosen to examine the robustness of the results for the sample from 2014 to 2021. Fourth, considering the effect of variable selection, the monthly average of temperature 2 meters above the surface (Temperature_2m) from the same database was selected to replace the core independent variable for the regression analysis. Fifth, considering the alternative measures of systematic risk of the explanatory variables, the choice of replacement of the explanatory variables selected the centrality of network topology features. This means that the Degree variable replaced the centrality of feature vectors for regression. Sixth, considering the sensitivity of the test results to different model scales, this paper resets the thresholds of network filtering. It selects 0.3 and 0.5 quartile thresholds to extract adequate information on the network obtained by Pearson's correlation coefficient.

Table 6 shows the seven robustness tests in columns (1)-(7), considering removing macro control variables, winsorizing, narrowing the sample interval and replacing the core explanatory variables, replacing the explanatory variables, and modifying the model scales, respectively. As can be seen from the results of removing the macro control variables in column (1) and replacing the explanatory variables in column (4) of Table 6, the goodness of fit of the model is reduced compared to the two-way fixed effects of including the micro-macro control variables. However, the effects of the variables remain unchanged and do not affect the overall robustness tests. The panel estimation results show that the variables maintain the same significance and direction of action as in the benchmark regression, so the conclusions are relatively robust.

## Results and analysis of the dynamic evolutionary patterns of temperature difference and the systemic risk in pan-financial market

### Impulse response analysis of higher and colder temperature and pan-financial systemic risk

Mean value panel regression shows the linear relationship between higher temperature, colder temperature, and pan-financial systemic risk. However, it cannot see how the three variables

**Table 6. Robustness test results.**

| | Dependent variable: systemic risk of pan-financial market | | | | | | |
|---|---|---|---|---|---|---|---|
| | (1) | (2) | (3) | (4) | (5) | (6) | (7) |
| | Remove macro | Winsorize | Narrow | Replace X | Replace Y | 0.3 quartiles | 0.5 quartiles |
| Temp40 | 0.0352*** | 0.0202** | 0.0321*** | 0.0232** | 3.5506*** | 0.0231*** | 0.0245*** |
| | (3.9847) | (2.2410) | (3.1432) | (2.3718) | (4.1513) | (2.7485) | (2.7258) |
| Constant | 0.5352*** | 0.0793*** | 0.8013*** | 0.4296*** | -7.4002 | 0.4195 | 0.5340 |
| | (2.9769) | (0.2143) | (2.9809) | (1.3379) | (-0.2420) | (1.3727) | (1.5650) |
| R-squared | 0.0333 | 0.0879 | 0.0899 | 0.0826 | 0.1472 | 0.0827 | 0.0822 |
| Temp80 | 0.0525*** | 0.0369** | 0.0589*** | 0.0399** | 7.0841*** | 0.0423*** | 0.0460*** |
| | (3.2610) | (2.2256) | (3.0483) | (2.1918) | (4.2105) | (2.7027) | (2.7557) |
| Constant | 0.5306*** | 0.0767** | 0.8031*** | 0.4276*** | -7.2942 | 0.4180 | 0.5334 |
| | (2.9530) | (0.2075) | (2.9866) | (1.3322) | (-0.2383) | (1.3674) | (1.5627) |
| R-squared | 0.0329 | 0.0879 | 0.0899 | 0.0825 | 0.1474 | 0.0827 | 0.0822 |
| Temp120 | 0.0759*** | 0.0608** | 0.0938*** | 0.0668** | 10.9342*** | 0.0701*** | 0.0770*** |
| | (3.0839) | (2.4275) | (3.1782) | (2.4516) | (4.1966) | (2.9477) | (3.0371) |
| Constant | 0.5296*** | 0.0805** | 0.8102*** | 0.4336*** | -6.9808 | 0.4238 | 0.5401 |
| | (2.9500) | (0.2178) | (3.0107) | (1.3496) | (-0.2279) | (1.3851) | (1.5811) |
| R-squared | 0.0328 | 0.0880 | 0.0899 | 0.0826 | 0.1474 | 0.0828 | 0.0823 |
| Temp_plus40 | 0.0361** | 0.0428*** | 0.0499*** | 0.0321* | 4.9595*** | 0.0394** | 0.0373** |
| | (2.1302) | (2.7124) | (2.8890) | (1.9292) | (3.2176) | (2.5291) | (2.2518) |
| Temp_minus40 | -0.0344*** | -0.0033** | -0.0150** | -0.0143* | -2.1232 | -0.0068 | -0.0117 |
| | (-3.0296) | (0.2532) | (-1.0202) | (-1.0400) | (-1.3934) | (-0.5656) | (-0.9279) |
| Constant | 0.5347*** | 0.0654*** | 0.7918*** | 0.4234** | -8.8392 | 0.4062 | 0.5235 |
| | (2.0071) | (0.1772) | (2.9433) | (1.3187) | (-0.2877) | (1.3328) | (1.5385) |
| R-squared | 0.0333 | 0.0881 | 0.0900 | 0.0826 | 0.1473 | 0.0828 | 0.0823 |
| Temp_plus80 | 0.0408* | 0.0553* | 0.0789** | 0.0454* | 7.2147** | 0.0565* | 0.0536* |
| | (1.3151) | (1.8876) | (2.4919) | (1.4458) | (2.5059) | (1.9439) | (1.7275) |
| Temp_minus80 | -0.0635*** | -0.0180** | -0.0394* | -0.0345* | -6.9527** | -0.0282 | -0.0385 |
| | (-2.9110) | (-0.7605) | (-1.4348) | (-1.3192) | (-2.4525) | (-1.2412) | (-1.6168) |
| Constant | 0.5342*** | 0.0708*** | 0.7973*** | 0.4254*** | -7.3554 | 0.4117 | 0.5300 |
| | (2.3360) | (0.1919) | (2.9661) | (1.3274) | (-0.2393) | (1.3520) | (1.5588) |
| R-squared | 0.0329 | 0.0879 | 0.0899 | 0.0825 | 0.1474 | 0.0827 | 0.0822 |
| Temp_plus120 | 0.0478* | 0.0724** | 0.1077** | 0.0595* | 6.6156 | 0.0769* | 0.0730 |
| | (1.0218) | (1.6461) | (2.2840) | (1.2667) | (1.5299) | (1.7524) | (1.5596) |
| Temp_minus120 | -0.1023*** | -0.0488** | -0.0801* | -0.0740* | -15.2538*** | -0.0633* | -0.0810** |
| | (-3.0072) | (-1.3575) | (-1.9136) | (-1.8674) | (-3.3968) | (-1.8249) | (-2.2252) |
| Constant | 0.5352*** | 0.0784*** | 0.8076*** | 0.4356*** | -6.0026 | 0.4218 | 0.5412 |
| | (2.3787) | (0.2122) | (2.9983) | (1.3552) | (-0.1950) | (1.3815) | (1.5874) |
| R-squared | 0.0328 | 0.0880 | 0.0899 | 0.0826 | 0.1474 | 0.0828 | 0.0823 |
| Control Variables | YES | YES | YES | YES | YES | YES | YES |
| Individual Effect | YES | YES | YES | YES | YES | YES | YES |
| Time Effect | YES | YES | YES | YES | YES | YES | YES |
| Observations | 12,320 | 12,320 | 10,416 | 12,320 | 12,320 | 12,320 | 12,320 |
| Number Of Firms | 112 | 112 | 112 | 112 | 112 | 112 | 112 |

Note

***, ** and *denote significance at 1%, 5% and 10% levels, respectively; t-statistics in parentheses.

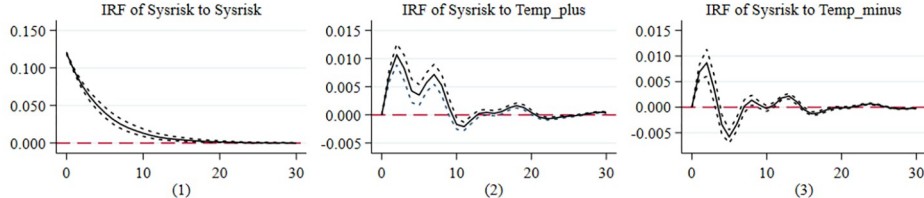

**Fig 3. Impulse response.**

change over time, which could make the causal logic chain hard to follow. Therefore, we choose to construct panel vector autoregression (PVAR) models of higher temperature, colder temperature, and systemic risk of pan-financial market and carry out impulse response (IRF) analysis in order to more intuitively portray the trend of the influence of higher temperature and colder temperature on the systemic risk of pan-financial market and study their dynamic time-varying evolution law. The impulse response is the dynamic effect of a shock of one standard deviation applied to the error term of one variable on the current and future of another variable, assuming all other factors are held constant. In the paper, we estimate a second-order panel vector autoregressive model and perform 1000 Monte-Carlo simulations with lag intervals of 0–30 periods. The impulse responses obtained between the variables are shown in Fig 3. The horizontal axis is the number of lags, and the vertical axis is the magnitude of the impulse response.

In Fig 3, subplots (2)-(3) show the impulse response relationships of higher temperature on systemic risk of pan-financial market and colder temperature on systemic risk of pan-financial market, respectively. Subfigure (2) shows that when pan-financial systemic risk is faced with the shock of higher temperature, the higher temperature has a particular positive effect on pan-financial systemic risk. The duration of this effect is around nine months, which indicates a time lag effect in exacerbating systemic risk by higher temperature, probably because the market will give feedback on the higher temperature only after the higher temperature has reached a certain level. The firm's production and operations are disrupted, resulting in loan delinquency, affecting the entire pan-financial market and exacerbating the firm's systemic risk. Subfigure (3) shows a positive shock effect of colder temperature on pan-financial systemic risk for about four months, which is shorter than the higher temperature period, and then becomes negative and tends to flatten out gradually. The possible explanation lies in the fact that the paper's systemic risk measurement uses stock data, and in the short term, investors will have a synergistic effect when the temperature changes and collectively sell climate-sensitive assets on a large scale, which will result in stock market shocks and exacerbate systemic risk, but in the long run, colder temperature favors business production and operation, thus reducing the systemic risk of pan-financial market. This impulse response further validates the mechanism of the relationship among the three. Also, based on the comparison of subfigures (2) and (3), it is found that higher temperature plays a more significant role and has a more substantial impact on the systemic risk of pan-financial market.

## Variance decomposition of higher and colder temperature and pan-financial systemic risk

The variance decomposition can measure the contribution of different disturbance terms to the volatility of endogenous variables and more accurately examine the magnitude of the interaction among pan-financial systemic risk's higher and colder temperature. The variance

**Table 7. Variance decomposition results for systemic risk in the traditional financial sectors.**

| Sector | Response Variable | Phase | Impulse Variable | | | | | |
|---|---|---|---|---|---|---|---|---|
| | | | Traditional financial market | | | Pan-financial market | | |
| | | | Sysrisk | Temp_plus | Temp_minus | Sysrisk | Temp_plus | Temp_minus |
| Banking | Sysrisk | 1 | 1.000 | 0.000 | 0.000 | 1.000 | 0.000 | 0.000 |
| | | 10 | 0.974 | 0.015 | 0.011 | 0.71 | 0.166 | 0.124 |
| | | 20 | 0.974 | 0.015 | 0.011 | 0.653 | 0.202 | 0.144 |
| | | 30 | 0.974 | 0.015 | 0.011 | 0.637 | 0.213 | 0.15 |
| Insurance | Sysrisk | 1 | 1.000 | 0.000 | 0.000 | 1.000 | 0.000 | 0.000 |
| | | 10 | 0.988 | 0.003 | 0.008 | 0.989 | 0.006 | 0.006 |
| | | 20 | 0.988 | 0.003 | 0.009 | 0.987 | 0.007 | 0.006 |
| | | 30 | 0.988 | 0.003 | 0.009 | 0.987 | 0.007 | 0.006 |
| Securities | Sysrisk | 1 | 1.000 | 0.000 | 0.000 | 1.000 | 0.000 | 0.000 |
| | | 10 | 0.98 | 0.003 | 0.017 | 0.904 | 0.056 | 0.04 |
| | | 20 | 0.979 | 0.004 | 0.017 | 0.894 | 0.062 | 0.044 |
| | | 30 | 0.979 | 0.004 | 0.017 | 0.893 | 0.063 | 0.044 |

Note: Due to space constraints, only the results of issues 1, 10, 20, and 30 are reported; if you need the full results, please contact the authors for a copy.

decomposition results can be obtained simultaneously during the impulse response analysis, and the number of analysis periods is set to 30.

The results of the variance decomposition of systemic risk in the traditional financial sector (banking, insurance, and securities) are shown in Table 7. A comparison of the two columns of Table 7, which show the variance decomposition of systemic risk with the three sectors of traditional finance not included and included in pan-financial market, shows that each sector's systemic risk contributes the most to itself, with the contributions of higher and colder temperature increasing from period to period. Compared to the traditional financial market, higher and colder temperature is more significant in pan-financial market for the securities and banking sectors, indicating that changes in systemic risk in the securities and banking sectors from temperature difference over the long run have a more significant impact. This finding is consistent with the empirical results of the fixed effects model above. Since temperature difference contributes more to systemic risk across sectors within pan-financial market, the predictive accuracy of the impact of temperature difference on systemic risk can be improved by establishing early warning mechanisms in pan-financial market.

The results of the variance decomposition of pan-financial systemic risk and its sectors are shown in Table 8. pan-financial systemic risks contribute to themselves more than other variables in the whole sample and across sectors. This suggests that pan-financial systemic risk has some self-reinforcing mechanisms in the whole sample and across sectors. In terms of the contribution of the variables to each other, higher and colder temperature affects systemic risk in the securities and banking sectors to a greater extent than in other sectors, which suggests that systemic risk of pan-financial market in the securities and banking sectors is more dependent on changes in temperature, a finding that is consistent with the empirical results of the fixed-effects model above and which further confirms the robustness of the findings of this paper.

## Identifying potential pathways for the transmission of temperature difference and systemic risk in pan-financial market

Based on the above mean panel regression and dynamic evolution law of temperature difference and systemic risk of pan-financial market, the paper further explores the potential

**Table 8. Results of the variance decomposition of systemic risk in pan-financial market and its sectors.**

| Sector | Response Variable | Phase | Impulse Variable | | |
|---|---|---|---|---|---|
| | | | Sysrisk | Temp_plus | Temp_minus |
| Pan-Financial Market | Sysrisk | 1 | 1.000 | 0.000 | 0.000 |
| | | 10 | 0.987 | 0.008 | 0.004 |
| | | 20 | 0.987 | 0.009 | 0.005 |
| | | 30 | 0.987 | 0.009 | 0.005 |
| Banking | Sysrisk | 1 | 1.000 | 0.000 | 0.000 |
| | | 10 | 0.710 | 0.166 | 0.124 |
| | | 20 | 0.653 | 0.202 | 0.144 |
| | | 30 | 0.637 | 0.213 | 0.15 |
| Insurance | Sysrisk | 1 | 1.000 | 0.000 | 0.000 |
| | | 10 | 0.989 | 0.006 | 0.006 |
| | | 20 | 0.987 | 0.007 | 0.006 |
| | | 30 | 0.987 | 0.007 | 0.006 |
| Security | Sysrisk | 1 | 1.000 | 0.000 | 0.000 |
| | | 10 | 0.904 | 0.056 | 0.040 |
| | | 20 | 0.894 | 0.062 | 0.044 |
| | | 30 | 0.893 | 0.063 | 0.044 |
| Real Estate | Sysrisk | 1 | 1.000 | 0.000 | 0.000 |
| | | 10 | 0.992 | 0.003 | 0.004 |
| | | 20 | 0.992 | 0.004 | 0.004 |
| | | 30 | 0.992 | 0.004 | 0.004 |
| New Energy | Sysrisk | 1 | 1.000 | 0.000 | 0.000 |
| | | 10 | 0.980 | 0.013 | 0.006 |
| | | 20 | 0.981 | 0.013 | 0.006 |
| | | 30 | 0.981 | 0.013 | 0.006 |
| Traditional Energy | Sysrisk | 1 | 1.000 | 0.000 | 0.000 |
| | | 10 | 0.974 | 0.020 | 0.006 |
| | | 20 | 0.973 | 0.020 | 0.006 |
| | | 30 | 0.973 | 0.020 | 0.007 |

Note: Due to space constraints, only the results of issues 1, 10, 20, and 30 are reported; if you need the full results, please contact the authors for a copy.

transmission pathways among higher temperature and colder temperature and systemic risk in pan-financial market. Therefore, the paper constructs a risk spillover network of temperature difference and systemic risk of pan-financial market sectors based on the TVP-VAR-DY framework and utilizes the MST algorithm to identify the minimum spanning tree of this network to discover the evolution pattern of the sectors at the core of the network as well as the paths of conduction.

As one of the main driving elements of the structural changes in the economic and financial system, climate change has the typical characteristics of "long-term, structural and systemic" [10]. Climate change may lead to a decline in the value of collaterals and stricter credit conditions, which may affect asset prices and credit conditions of less risk-resistant firms due to the financial accelerator effect and financial constraints, thus spreading their risks along the financial network to other sectors. Constraints, asset prices, and credit conditions of less risk-resistant firms will be affected, thereby spreading their risks to other sectors along financial networks, and market signals may amplify the severity of climate change risks so that the impact of climate change on a single financial institution may evolve into a systemic risk. The

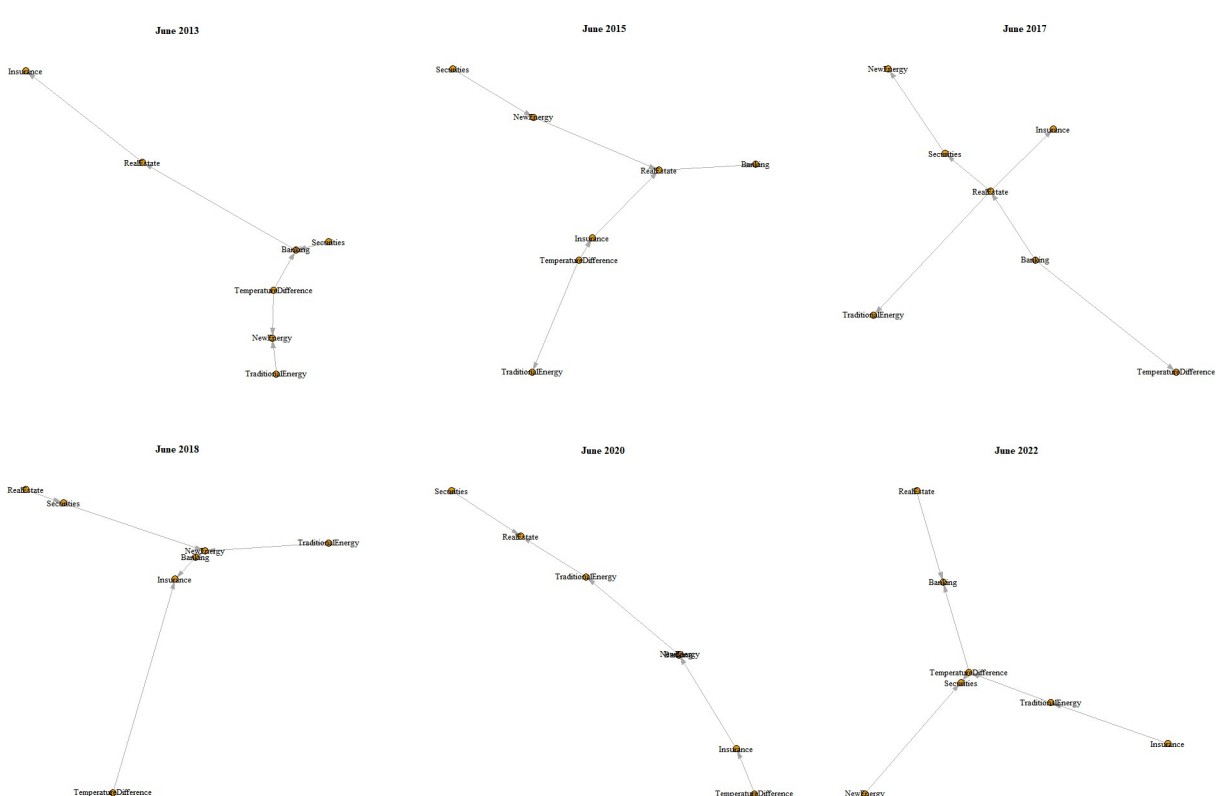

**Fig 4. Minimum spanning tree network for systemic risk and temperature difference across sectors in pan-financial market.**

paper utilizes the Minimum Spanning Tree (MST) algorithm to capture the shortest contagion path for systemic risk contagion within a network. Due to space limitations, six subgraphs in Fig 4 show the shortest paths of the systemic risk and temperature difference contagion networks of different parts of pan-financial market in 2013, 2015, 2017, 2018, 2020, and 2022. The changes in the absolute core position in the network show the pattern of "banking-real estate-new energy-temperature difference," which confirms that the core position sectors will show different changes at different stages of economic development, with the most significant influence and potential damage. Possible explanations are: First, the release of the "Golden Ten" [the 2013 General Office of the State Council officially released the "Guiding Opinions on Financial Support for Economic Structural Adjustment and Transformation and Upgrading" (referred to as the "Golden Ten")], which calls for a better play of the role of financial support for economic restructuring, transformation, and upgrading, while with the continuous development of the bull market, banks, as the largest financial institutions supporting the development of the entity, naturally become risk takers, and the systemic risk of the securities sector is transmitted to the banking sector through the minimum spanning tree network. Second, from 2015 to 2017, the real estate inventory at the same time China's central government shouted developers "price reduction," the real estate sector began to occupy the core of the network, banks, and the real estate sector inextricably linked to the risk of mutual contagion between the two. Third, from 2018 to 2020, with the outbreak of the trade war between China and the U.S., the risk volatility between the energy and financial market has intensified. At the same time, in order to carry out the spirit of the 19th CPC National Congress, the CPC Central Committee has taken the main line of the supply-side structural reform to promote the energy

transformation in the new era and vigorously developed the new energy by many measures, in which the new energy occupies a central position. Fourth, according to the United Nations' World Meteorological Organization's State of the Global Climate Report, the global average annual temperature in 2021 is rising, the temperature difference is gradually occupying a central position in the network, and it is more likely to be linked to systemic risk spillovers in other sectors of pan-financial market, which is related to the stability of the overall network. The distance between the temperature difference and the securities sector is the shortest, and the temperature difference is associated with systemic risk spillovers in the securities sector. Suppose the sector is at the center node of the network. In that case, the risk will spread outward quickly along the MST, eventually affecting the entire pan-financial network, accelerating the contagion of systemic risk, and causing an economic crisis. At the same time, the temperature difference is at the network's core at this stage, which is closely related to the spillover of systemic risk among various sectors. We should concentrate on preventing the systemic risk that the temperature difference brings about because it plays a crucial role in the information transmission and stability of the entire network.

## Conclusions and suggestion

By constructing a pan-financial market volatility spillover network, the paper analyzes the relationship between temperature difference and systemic risk as well as the linkages between macroeconomic factors and firms' micro characteristics based on measuring systemic risk in China's pan-financial market and draws the following main conclusions: First, the systemic risk of China's pan-financial market has fluctuated significantly during significant events such as the 2013 bull market, the 2015 stock market crash, the 2016 A-share market circuit breaker, the 2017 de-leveraging, the 2018 U.S.-China trade friction, and the inclusion of the Corona Virus Disease 2019 (COVID-19), all of which are relatively consistent with the actual situation. Second, systemic risk in the traditional financial sector is more negligible in the financial pan-financial scenario than in the non-financial pan-financial scenario. However, the diffusion of systemic financial risk has expanded with the increased pan-financial market. Third, the temperature difference is significantly and positively related to systemic risk of China's pan-financial market, which means that an increase in temperature difference exacerbates the systemic risk of pan-financial market; the degree of exacerbation of systemic risk by higher temperature is significantly greater than the degree of suppression of systemic risk by colder temperature, which means that higher and colder temperature has a significant asymmetric effect on the systemic risk of pan-financial market; and, as further empirically examined from the sectoral level, higher temperature significantly exacerbates systemic risk in the banking sector, and decreasing temperature significantly suppresses systemic risk in securities and new energy, suggesting that there is sectoral heterogeneity in the impact of temperature difference on systemic risk. Fourth, macroeconomic factors and firm micro characteristics have some explanatory power for systemic risk and show some differences among the six sectors of pan-financial market. Specifically, macroeconomic variables, except the China real estate climate index, positively affect the systemic risk of pan-financial market; firm micro-level scale positively affects the systemic risk of pan-financial market; turnover and leverage negatively affect the systemic risk of pan-financial market. This further suggests that temperature difference significantly affects systemic risk, which is also influenced by macroeconomic and firm micro-level control variables. Fifth, in terms of dynamic evolution characteristics, based on the impulse response analysis of pan-financial systemic risk and temperature difference, the higher temperature will exacerbate systemic risk. Colder temperature will contribute to financial systemic risk in the short term. However, it will be suppressed in the long term and gradually weaken and

eventually disappear, and the degree of influence of higher temperature is greater than that of colder temperature. The results of the variance decomposition analysis indicate that higher and colder temperature contributes more to the variance of systemic risk in the banking and securities sectors of pan-financial market and have a more substantial exacerbating effect on systemic risk, which should be focused on these two sectors. Sixth, from the perspective of the potential path of systemic risk transmission between temperature difference and pan-financial market, based on the MST network, it is found that the temperature difference is closely linked with the six sectors of pan-financial market systemically, and the evolution of the core position is "Bank→Real Estate→New Energy→Temperature Difference." After 2021, the temperature difference takes the center position in the network, and the shock on the systemic risk of each sector of pan-financial market increases. Furthermore, the systemic risk will be transmitted rapidly along the minimum spanning tree network when there is an external shock.

Uncertainty brought about by climate shocks exacerbates the trend of pan-financial market in traditional financial market, profoundly changing the intensity, scope of impact, and path of contagion of systemic financial risks, which provides essential insights into current systemic financial risks:

First, promote cooperation among firms in pan-financial market. Although the insurance, energy, and real estate sectors are less affected by higher and colder temperature, their systemic risks are closely linked to the industries, so promoting cooperation among the sectors should be strengthened. The overall reduction of systemic risks can be achieved through the integration of risks, thus contributing to the optimization of the industry structure of pan-financial market and the reduction of the impact of the climate shocks on pan-financial market's systemic risks. Strengthening cooperation within pan-financial market can be carried out in the following aspects for example, constructing a unified systemic risk management system, sharing data on climate change shocks in various sectors, improving the distribution of the industry chain structure of the respective sectors with climate-related systemic risks; and carrying out business diversification and cooperation in lending to climate transition-sensitive sectors.

Second, it strengthens investors' ability to assess and manage climate risk. Climate change is an essential source of financial systemic risk, although climate change risk has only attracted attention in recent years. The reaction of stock prices to climate change provides a signal to investors and encourages relevant institutions and investors to carry out adequate monitoring of climate information, risk analysis taking climate-related variables into account, and planning for climate change can improve the accuracy of predictions of systemic risk, which is conducive to investors' investment decisions, the stable development of the stock market, and the sustained and stable growth of China's macroeconomy.

Third, policymakers should consider climate change factors. Impacts from temperature difference can generate intersectoral contagion along the shortest path (from the minimum spanning tree) and have unforeseen and potentially large-scale severe consequences for the stability of pan-financial market. Policymakers should consider temperature shocks as an indicator of financial systemic risk and be concerned about the potential for systemic risks and crises at higher temperature. Policymakers may also promote a green transformation of pan-financial system to maintain financial stability in climate change. For example, financial institutions could consider green financial instruments when providing loans and advances to real businesses, emphasizing environmental and social responsibility, focusing on this potential channel of temperature impacts on pan-financial systemic risk, and mitigating the negative impacts of climate change-induced systemic risk to maintain financial stability.

Fourthly, Governments should expand the scenarios for monetary policy and macroprudential regulation. As climate policy can effectively curb the greenhouse effect and contribute to the green and low-carbon transformation of the economy, it may also adversely affect

macroeconomic and financial stability and induce pan-financial systemic risks due to climate change. Therefore, it is recommended that the relevant departments of China jointly research climate policy programs, learn from the practical experience of the international community, explore the feasibility, and strengthen the incentive effect. On the other hand, it is recommended to pay great attention to the systemic risks that the implementation of climate policies may trigger and fully pre-assess the impacts of the implementation of stringent climate policies on the economy and society, especially the impacts of temperature difference on the stability of pan-financial system in this context, and carry out stress tests. Most importantly, considering the strong complementarity between climate governance policies and macroprudential financial regulation, it is recommended that the financial sector, central banks, and financial regulators further strengthen their communication and collaboration, incorporate climate change into the two-pillar regulatory framework, strengthen the coordination of the two-pillar regulation of climate, monetary and macroprudential policies, and flexibly utilize them according to their respective strengths, to fully unleash the governance effects of climate policies and effectively avoid risks at the same time. Consideration can also be given to drawing on the experience of developed economies in climate governance to promote the linkage of climate policy with financial instruments, such as insurance and credit, to maximize the policy effect of dual-pillar regulation and control.**References**

## Supporting information

**S1 Appendix. Data file.**
(XLSX)

## Author Contributions

**Conceptualization:** Kaiwei Jia.

**Data curation:** Yunqing Du.

**Investigation:** Yunqing Du.

**Methodology:** Kaiwei Jia.

**Resources:** Kaiwei Jia.

**Software:** Yunqing Du.

**Validation:** Kaiwei Jia.

**Visualization:** Yunqing Du.

**Writing – original draft:** Yunqing Du.

**Writing – review & editing:** Kaiwei Jia.

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

**31.** Brunnermeier MK, Sockin M, Xiong W. China's Model of Managing the Financial System. Review of Economic Studies. 2022; 89(6):3115–53.https://doi.org/10.1093/restud/rdab098.

**32.** Yang Z, Chen Y, Huang Z. Research on Influencing Factors and Transmission Channels of Systemic Risks under International Shocks.Economic Research Journal.2023, 58(01):90–106.

**33.** Olabisi M. Input-Output Linkages and Sectoral Volatility. Economica. 2020; 87(347):713–46.https://doi.org/10.1111/ecca.12327.

**34.** He X, Hamori S. Is volatility spillover enough for investor decisions? A new viewpoint from higher moments. Journal of International Money and Finance. 2021;116.https://doi.org/10.1016/j.jimonfin.2021.102412.

**35.** Brunnermeier M, Rother S, Schnabel I. Asset Price Bubbles and Systemic Risk. Review of Financial Studies. 2020; 33(9):4272–317.https://doi.org/10.1093/rfs/hhaa011.

**36.** Biswas S, Kumar R. Bank board network and financial stability in emerging markets. Emerging Markets Review. 2022;51.https://doi.org/10.1016/j.ememar.2022.100884.

**37.** Chen M, Huang D, Wu B. Interlocking Directorates and Firm Performance: Evidence from China. Social Science Electronic Publishing. https://doi.org/10.2139/ssrn.4005022.

**38.** Billio M, Getmansky M, Lo AW, Pelizzon L. Econometric measures of connectedness and systemic risk in the finance and insurance sectors. Journal of Financial Economics. 2012; 104(3):535–59.https://doi.org/10.1016/j.jfineco.2011.12.010.

**39.** Hardie WK, Wang WN, Yu LN. TENET: Tail-Event driven NETwork risk. Journal of Econometrics. 2016; 192(2):499–513.https://doi.org/10.1016/j.jeconom.2016.02.013.

**40.** Demirer M, Diebold FX, Liu L, Yilmaz K. Estimating global bank network connectedness. Journal of Applied Econometrics. 2018; 33(1):1–15.https://doi.org/10.2139/ssrn.2631479.

**41.** Wang GJ, Chen YY, Si HB, Xie C, Chevallier J. Multilayer information spillover networks analysis of China's financial institutions based on variance decompositions. International Review of Economics & Finance. 2021; 73:325–47.https://doi.org/10.1016/j.iref.2021.01.005.

**42.** Diebold FX, Yilmaz K. On the network topology of variance decompositions: Measuring the connectedness of financial firms. Journal of Econometrics. 2014; 182(1):119–34.https://doi.org/10.1016/j.jeconom.2014.04.012.

**43.** Fang Y,Shao ZQ.The Influence of Social Security Fund's Stock Investment on Stock Market Volatility Risk: Evidence from China. Modern Economic Science. 2022; 44(04):59–75.

**44.** Kumar SK, Madhu. Temperature and production efficiency growth: empirical evidence. Clim Change. 2019; 156(1a2).https://doi.org/10.1007/s10584-019-02515-5.

**45.** Pan M, Liu H, Cheng Z.The impact of Extreme Climate on Commercial Banks' Risk-taking: Evidence from Local Commercial Banks in China.Journal of Financial Research.2023; 10:39–57.

**46.** Song XN, Fang T. Temperature shocks and bank systemic risk: Evidence from China. Finance Research Letters. 2023;51.https://doi.org/10.1016/j.frl.2022.103447.

**47.** Nicholson WB, Matteson DS, Bien J. VARX-L: Structured regularization for large vector autoregressions with exogenous variables. International Journal of Forecasting. 2017; 33(3):627–51.https://doi.org/10.1016/j.ijforecast.2017.01.003.

**48.** Gong XL,Xiong X,Zhang W.Research on Systemic Risk Measurement and Spillover Effect of Financial Institutions in China. Journal of Management World. 2020; 36(08):65–83.

**49.** Zhu B,Ma YT.Sectoral Characteristics, Monetary Policy and Systemic Risk—An Analysis Based on the "Economic and Financial" Linkage Network.Studies of International Finance. 2018;(04):22–32.

**50.** Zhang WP,Zhuang XT, Wang J.Systematic Risk Spatial Spillover Correlation and Risk Prediction Analysis of Cross-industry in China' Stock Market—Based on The Tail Risk Network Model.Chinese Journal of Management Science. 2021; 29(12):15–28.

**51.** Newman MEJ. The structure and function of complex networks. Siam Review. 2003; 45(2):167–256. https://doi.org/10.1137/S003614450342480.

**52.** Guo WW.Structural Deleveraging and Systematic Risk Spillover inFinancial Institutions: Promoting or Suppressing?. Journal of Central University of Finance & Economics. 2020;(04):26–41.

**53.** Adrian T, Brunnermeier MK. CoVaR. American Economic Review. 2016; 106(7):1705–41.https://doi.org/10.1257/aer.20120555.

**54.** Aevoae GM, Andrieş AM, Ongena S, Sprincean N. ESG and systemic risk. Appl Econ. 2022; 55 (27):3085–109.https://doi.org/10.1080/00036846.2022.2108752.

**55.** Kahn ME, Mohaddes K, Ng RNC, Pesaran MH, Raissi M, Yang JC. Long-term macroeconomic effects of climate change: A cross-country analysis. Energy Economics. 2021;104.https://doi.org/10.1016/j.eneco.2021.105624.

**56.** Barigozzi M, Brownlees C. NETS: Network estimation for time series. Journal of Applied Econometrics. 2019; 34(3):347–64. https://doi.org/10.1002/jae.2676.

**57.** Cevik S, Jalles JT. This changes everything: Climate shocks and sovereign bonds. Energy Econ. 2022; 107:105856. https://doi.org/10.1016/j.eneco.2022.105856.

**58.** Dell M, Jones BF, Olken BA. Temperature Shocks and Economic Growth: Evidence from the Last Half Century. Am Econ J Macroecon. 2012; 4(3):66–95. https://doi.org/10.1257/mac.4.3.66.

**59.** Hsiang S. M. Temperatures and cyclones strongly associated with economic production in the Caribbean and Central America. Proceedings of the National Academy of Sciences of the United States of America. 2010; 107(35):p.15367–72. https://doi.org/10.1073/pnas.1009510107 PMID: 20713696

**60.** International Monetary Fund. Physical Risk and Equity Prices,Global Financial Stability Report, Chapter 5  Washington, DC:  International Monetary Fund.2020.

**61.** Liu Y, Huang C, Zou Z, Chen Q, Chu X. Research into the Mechanism for the Impact of Climate Change on Systemic Risk—A Case Study of China's Small- and Medium-sized Commercial Banks. Sustainability. 2020; 12(22):9582. https://doi.org/10.3390/su12229582.

**62.** PéREZ-GONZáLEZ F, Yun H. Risk Management and Firm Value: Evidence from Weather Derivatives. The Journal of Finance. 2013; 68(5):2143–76. https://doi.org/10.1111/jofi.12061.

