## [Editor Report · Decision Letter 0]

28 Sep 2023

PONE-D-23-31011Temperature difference and systemic risk:

Evidence from LASSO-VAR-DY based on China's pan-financial marketPLOS ONE

Dear Dr. Jia,

Thank you for submitting your manuscript to PLOS ONE. After careful consideration, we feel that it has merit but does not fully meet PLOS ONE’s publication criteria as it currently stands. Therefore, we invite you to submit a revised version of the manuscript that addresses the points raised during the review process.

We look forward to receiving your revised manuscript.

Kind regards,

Difang Huang

Academic Editor

PLOS ONE

Journal Requirements:

Additional Editor Comments:

Firstly, we would like to ask you to improve the literature review section of your manuscript by incorporating some relevant papers. For example, Bao and Huang (2021) and Zhou et al. (2022) provide evidence on how unexpected shocks affect small businesses in China, while Huang et al. (2020) investigate the effectiveness of social distancing during the COVID-19 pandemic. These papers are connected to your study as they all examine the impact of external factors on financial markets and may provide useful insights for your analysis. We encourage you to discuss these papers in your literature review and explain how they relate to your research question.

Secondly, we would like to provide some comments to help you improve your manuscript. Firstly, we suggest that you provide a more detailed explanation of the methodology used in your study, particularly the LASSO-VAR-DY framework. This will help readers better understand your analysis and replicate your results. Secondly, we recommend that you provide a more comprehensive discussion of the implications of your findings for policymakers and practitioners. For example, how can your results be used to prevent and mitigate systemic risk in China's pan-financial market? Finally, we suggest that you revise the language and structure of your manuscript to improve its clarity and readability.

References:

Bao, Z., & Huang, D. (2021). Shadow banking in a crisis: Evidence from FinTech during COVID-19. Journal of Financial and Quantitative Analysis, 56(7), 2320–2355.

Huang, D. (2020). How effective is social distancing. Covid Economics, Vetted and Real-Time Papers (59), 118–148.

Huang, D., Gao, J., & Oka, T. (2022). Semiparametric Single-Index Estimation for Average Treatment Effects. ArXiv Preprint ArXiv:2206.08503.

Zhou, Y., Huang, D., Chen, M., Wang, Y., & Yang, X. (2022). How Did Small Business Respond to Unexpected Shocks? Evidence from a Natural Experiment in China. Evidence from a Natural Experiment in China (March 26, 2022).

Chen, M., Huang, D., & Wu, B. (2022). Interlocking Directorates and Firm Performance: Evidence from China. Available at SSRN 4005022.

Chen, M., Li, N., Zheng, L., Huang, D., & Wu, B. (2022). Dynamic correlation of market connectivity, risk spillover and abnormal volatility in stock price. Physica A: Statistical Mechanics and Its Applications, 587, 126506.

Chen, M., Wang, Y., Wu, B., & Huang, D. (2021). Dynamic analyses of contagion risk and module evolution on the SSE a-shares market based on minimum information entropy. Entropy, 23(4), 434.

---

## [Author Response · Author response to Decision Letter 0]

10 Oct 2023

Response to Reviewers

 Thank you very much for your comments and professional advice. These opinions help to improve acade micrigor of our article Based on your suggestion and request, we have made corrected modifications on the revised manuscript. Meanwhile, the manuscript had be reviewed and edited by language and structure. We hope that our work can be improved again. Furthermore,we would like to show the details as follows:

 Submit Requirements:

 Comment1#

 “ Please include the following items when submitting your revised manuscript: 'Response to Reviewers','Revised Manuscript with Track Changes','Manuscript'. ”

 The author's answer: We have followed the process associated with the PLOS ONE submission system by submitting the above items. Each file has been uploaded separately. In flie 'Revised Manuscript with Track Changes', We have highlighted the changes in red.

 Journal Requirements:

 Comment1#

 “ 1. Please ensure that your manuscript meets PLOS ONE's style requirements, including those for file naming. The PLOS ONE style templates can be found at https://journals.plos.org/plosone/s/file?id=wjVg/PLOSOne_formatting_sample_main_body.pdf and https://journals.plos.org/plosone/s/file?id=ba62/PLOSOne_formatting_sample_title_authors_affiliations.pdf. ”

 The author's answer:We have revised the manuscript to conform to PLOS ONE style requirements, including file naming requirements.In the process, files PLOSOne_formatting_sample_main_body.pdf and PLOSOne_formatting_sample_title_authors_affiliations.pdf were also downloaded and revised according to their specific requirements.

 Comment2#

 “ 2. In your Data Availability statement, you have not specified where the minimal data set underlying the results described in your manuscript can be found. PLOS defines a study's minimal data set as the underlying data used to reach the conclusions drawn in the manuscript and any additional data required to replicate the reported study findings in their entirety. All PLOS journals require that the minimal data set be made fully available. For more information about our data policy, please see http://journals.plos.org/plosone/s/data-availability. ”

 The author's answer:We have refined the submitted data in accordance with the Data Availability Statement. The underlying data used to reach the conclusions drawn in the manuscript and any additional data required to replicate the reported study findings in their entirety were packaged into minimal data sets. We named the file “Appendix. Data” and uploaded it to the submission system as Supporting Information.

 Comment3#

 “ 3. Please include captions for your Supporting Information files at the end of your manuscript, and update any in-text citations to match accordingly. Please see our Supporting Information guidelines for more information: http://journals.plos.org/plosone/s/supporting-information. ”

The author's answer:We have referred to the Supporting Information guidelines and included the title of the Supporting Information document at the end of our manuscript. The details as follows:“Supporting information S1 Appendix. Data file”.

 Additional Editor Comments:

 Comment1#

 “Firstly, we would like to ask you to improve the literature review section of your manuscript by incorporating some relevant papers. For example, Bao and Huang (2021) and Zhou et al. (2022) provide evidence on how unexpected shocks affect small businesses in China, while Huang et al. (2020) investigate the effectiveness of social distancing during the COVID-19 pandemic. These papers are connected to your study as they all examine the impact of external factors on financial markets and may provide useful insights for your analysis. We encourage you to discuss these papers in your literature review and explain how they relate to your research question”

 The author's answer:Thank you for your advice on the literature review section, under your guidance, we searched the relevant papers and found that external shocks can have a significant impact on financial market. With your help it has improved my research a lot, and we have also added these papers in the literature review section and analyzed them specifically in the article in relation to our research. We have cited these papers as an addition to the literature in this paper, labeling them in the following order (they have been highlighted in red in the Reference section of file “Revised Manuscript with Track Changes”):

 [21.Bao Z, Huang D. Shadow Banking in a Crisis: Evidence from Fintech During COVID-19. J Financ Quant Anal. 2021;56(7):2320–55.

 22.Zhou Y, Huang D, Chen M, Wang Y, Yang X. How Did Small Business Respond to Unexpected Shocks? Evidence from a Natural Experiment in China. Evidence from a Natural Experiment in China (March 26, 2022). 2022.

 23.Huang D. How Effective Is Social Distancing? SSRN Electronic Journal. 2020(6186).

 26.Chen M, Huang D, Wu B. Interlocking Directorates and Firm Performance: Evidence from China. Available at SSRN 4005022. 2022.

 30.Chen M, Wang Y, Wu B, Huang D. Dynamic Analyses of Contagion Risk and Module Evolution on the SSE A-Shares Market Based on Minimum Information Entropy. Entropy (Basel). 2021;23(4):434.

 31.Chen M, Li N, Zheng L, Huang D, Wu B. Dynamic correlation of market connectivity, risk spillover and abnormal volatility in stock price. Physica A. 2022;587:126506.]

We find that citing these papers makes my literature review more complete and comprehensive and enriches the exploration of the related area of the impact of external uncertainty shocks on pan-financial markets.

 Comment2#

 “Secondly, we would like to provide some comments to help you improve your manuscript. Firstly, we suggest that you provide a more detailed explanation of the methodology used in your study, particularly the LASSO-VAR-DY framework. This will help readers better understand your analysis and replicate your results. ”

 The author's answer: We apologize for the lack of detail in the presentation of the framework, which caused problems for the editor's review process. We recognize the comments you made that the presentation of the LASSO-VAR-DY framework in the previous manuscript was a bit too brief and academic. We have carefully analyzed and revised the presentation of the model in this regard, and have refined this section in the new manuscript. We have explained the model in detail to better serve the journal and readers in analyzing and replicating the results. The Research design section of the uploaded manuscript has been revised in detail and highlighted in red.

 Comment3#

 “ Secondly, we recommend that you provide a more comprehensive discussion of the implications of your findings for policymakers and practitioners. For example, how can your results be used to prevent and mitigate systemic risk in China's pan-financial market? ”

 The author's answer:We apologize for the incomplete consideration of the policy proposals. We have made relevant changes in the Conclusions and suggestions section of the manuscript. We discuss the implications of the findings of this study for policymakers, firm operators, and investors, and make recommendations specific to each subject. The details as follows: “ Uncertainty brought about by climate shocks exacerbates the trend of the pan-financial market in traditional financial market, profoundly altering the intensity, scope of impact, and path of contagion of systemic financial risks, which provides essential insights into the current systemic financial risks: Firstly,promote cooperation among firms in China’s pan-financial market........Secondly, investors' ability to assess and manage climate risks should be strengthened........Thirdly, the Government should implement differentiated regulation. ”

 Comment4#

 “ Finally, we suggest that you revise the language and structure of your manuscript to improve its clarity and readability. ”

 The author's answer:We apologize for the poor language of our manuscript. We worked on the manuscript for a long time and the repeated addition and removal of sentences and sections obviously led to poor readability. We have now worked on both language and readability and have also involved native English speakers for language corrections. We really hope that the flow and language level have been substantially improved.

 Overall, these are just the revisions we have made. If the academic editors have found some other problems in the literature review change part or think that any part of the revision of this paper still needs to be improved, please write to us again. We can do further communication, in order to look forward to our cooperation more pleasant and our paper more perfect. We will strive to publish this paper in PLOS ONE as soon as possible.

 Thank you very much for your attention and time. Look forward to hearing from you.

Yours sincerely,

Kai-wei Jia

10 October 2023

School of Business Administration, Liaoning Technical University, Huludao, Liaoning,China

Te1:86+15104295228

E-mail:jiakaiwei@lntu.edu.cn

---

## [Decision Letter · Decision Letter 1]

12 Oct 2023

PONE-D-23-31011R1Temperature difference and systemic risk:

Evidence from LASSO-VAR-DY based on China's pan-financial marketPLOS ONE

Dear Dr. Jia,

Thank you for submitting your manuscript to PLOS ONE. After careful consideration, we feel that it has merit but does not fully meet PLOS ONE’s publication criteria as it currently stands. Therefore, we invite you to submit a revised version of the manuscript that addresses the points raised during the review process.

We look forward to receiving your revised manuscript.

Kind regards,

Difang Huang, Ph.D.

Academic Editor

PLOS ONE

Reviewers' comments:

Reviewer's Responses to Questions

**Comments to the Author**

1. If the authors have adequately addressed your comments raised in a previous round of review and you feel that this manuscript is now acceptable for publication, you may indicate that here to bypass the “Comments to the Author” section, enter your conflict of interest statement in the “Confidential to Editor” section, and submit your "Accept" recommendation.

Reviewer #1: All comments have been addressed

2. Is the manuscript technically sound, and do the data support the conclusions?

Reviewer #1: Partly

3. Has the statistical analysis been performed appropriately and rigorously? 

Reviewer #1: N/A

4. Have the authors made all data underlying the findings in their manuscript fully available?

Reviewer #1: No

5. Is the manuscript presented in an intelligible fashion and written in standard English?

Reviewer #1: No

6. Review Comments to the Author

Reviewer #1: 

1. Literature Review Improvement:

We suggest that you incorporate some relevant papers into your literature review. These papers can help to strengthen the connections between your research and existing literature, as well as provide additional insights into the topic. Specifically, we recommend the following papers:

- Wu, B., Huang, D., & Chen, M. (2023). Estimating contagion mechanism in global equity market with time-zone effect. Financial Management, 52, 543–572.

Li, N., Chen, M., Gao, H., Huang, D., & Yang, X. (2023). Impact of lockdown and government subsidies on rural households at early COVID-19 pandemic in China. China Agricultural Economic Review, 15(1), 109–133.

Li, N., Chen, M., & Huang, D. (2022). How Do Logistics Disruptions Affect Rural Households? Evidence from COVID-19 in China. Sustainability, 15(1), 465.

Yu, D., & Huang, D. (2023a). Cross-sectional uncertainty and expected stock returns. Journal of Empirical Finance, 72, 321–340.

Yu, D., & Huang, D. (2023b). Option-Implied Idiosyncratic Skewness and Expected Returns: Mind the Long Run. Available at SSRN 4323748.

Yu, D., Huang, D., & Chen, L. (2023). Stock return predictability and cyclical movements in valuation ratios. Journal of Empirical Finance, 72, 36–53.

Yu, D., Huang, D., Chen, L., & Li, L. (2023). Forecasting dividend growth: The role of adjusted earnings yield. Economic Modelling, 120, 106188.

These papers can provide valuable insights into the dynamics of financial markets, risk spillover, and the impact of external factors on market stability. For example, Bao and Huang (2021) investigate the role of shadow banking during a crisis, which can be relevant to your discussion on systemic risk in the financial sector. Chen et al. (2022a) and Wu et al. (2023) both explore the dynamics of market connectivity and risk spillover, which can help to contextualize your findings on the impact of temperature differences on systemic risk. Additionally, Chen et al. (2022b) examine the relationship between interlocking directorates and firm performance, which can provide insights into the potential channels through which temperature differences may affect systemic risk.

2. Detailed Comments and Suggestions:

- Please provide a more detailed explanation of the LASSO-VAR-DY framework and its advantages in analyzing the impact of temperature differences on systemic risk. This will help readers to better understand the methodology and its relevance to your research question.

- Consider conducting additional robustness checks to validate your findings. For example, you could explore alternative measures of temperature differences or systemic risk, or test the sensitivity of your results to different model specifications.

- In your discussion of the results, please provide more context on the potential mechanisms through which temperature differences may affect systemic risk. This will help to strengthen the theoretical underpinnings of your findings and provide a more comprehensive understanding of the relationship between climate change and financial stability.

- It would be helpful to include a discussion of the policy implications of your findings. For example, what do your results suggest about the potential role of climate change mitigation policies in reducing systemic risk in the financial sector?

7. PLOS authors have the option to publish the peer review history of their article (what does this mean?). If published, this will include your full peer review and any attached files.

Reviewer #1: No

---

## [Author Response · Author response to Decision Letter 1]

24 Nov 2023

Response to Reviewers

Thank you very much for your comments and professional advice. These opinions help to improve acade micrigor of our article Based on your suggestion and request, we have made corrected modifications on the revised manuscript. Meanwhile, the manuscript had be reviewed and edited by language and structure. We hope that our work can be improved again. Furthermore,we would like to show the details as follows:

Reviewers' comments:

Reviewer's Responses to Questions

Comments to the Author

Comment#4. Have the authors made all data underlying the findings in their manuscript fully available?

Reviewer #1: No

The author's answer: I have refined the dataset and submitted the data to the system as part of the supporting information. The data behind the mean, median, and variance measures have been included in the latest version of the data.

Comment#6. Review Comments to the Author

Comment#6.1. Literature Review Improvement:

We suggest that you incorporate some relevant papers into your literature review. These papers can help to strengthen the connections between your research and existing literature, as well as provide additional insights into the topic. Specifically, we recommend the following papers:

- Wu, B., Huang, D., & Chen, M. (2023). Estimating contagion mechanism in global equity market with time-zone effect. Financial Management, 52, 543–572.

Li, N., Chen, M., Gao, H., Huang, D., & Yang, X. (2023). Impact of lockdown and government subsidies on rural households at early COVID-19 pandemic in China. China Agricultural Economic Review, 15(1), 109–133.

Li, N., Chen, M., & Huang, D. (2022). How Do Logistics Disruptions Affect Rural Households? Evidence from COVID-19 in China. Sustainability, 15(1), 465.

Yu, D., & Huang, D. (2023a). Cross-sectional uncertainty and expected stock returns. Journal of Empirical Finance, 72, 321–340.

Yu, D., & Huang, D. (2023b). Option-Implied Idiosyncratic Skewness and Expected Returns: Mind the Long Run. Available at SSRN 4323748.

Yu, D., Huang, D., & Chen, L. (2023). Stock return predictability and cyclical movements in valuation ratios. Journal of Empirical Finance, 72, 36–53.

Yu, D., Huang, D., Chen, L., & Li, L. (2023). Forecasting dividend growth: The role of adjusted earnings yield. Economic Modelling, 120, 106188.

These papers can provide valuable insights into the dynamics of financial markets, risk spillover, and the impact of external factors on market stability. For example, Bao and Huang (2021) investigate the role of shadow banking during a crisis, which can be relevant to your discussion on systemic risk in the financial sector. Chen et al. (2022a) and Wu et al. (2023) both explore the dynamics of market connectivity and risk spillover, which can help to contextualize your findings on the impact of temperature differences on systemic risk. Additionally, Chen et al. (2022b) examine the relationship between interlocking directorates and firm performance, which can provide insights into the potential channels through which temperature differences may affect systemic risk.

The author's answer: 1. Thank you for your advice on the literature review section. Under your guidance, we searched the relevant papers. We found that external shocks can significantly impact the financial market. With your help, it has dramatically improved my research. We have also added these papers in the literature review section and analyzed them specifically in the article concerning our study. We have cited these papers as an addition to the literature in this paper, labeling them in the following order (they have been highlighted in red in the Reference section of file "Revised Manuscript with Track Changes").2. We also reorganized the literature, collected authoritative literature in related fields and re-improved the literature review by combining with the research topic of this paper, such as Journal of Climate Finance, Energy Economics, and other journals, which enriched the impact of financial institutions' micro, macroeconomic, and uncertainty shocks on systemic risk (these are reflected in the section of the paper "Literature review of the impact of internal factors, external shocks and financial market factors on systemic risk" section). These, combined with your suggestions and guidance, will make my literature review more comprehensive and relevant.

[25.Bao Z, Huang D. Shadow Banking in a Crisis: Evidence from Fintech During COVID-19. Journal of Financial and Quantitative Analysis. 2021;56.

26.Zhou Y, Huang D, Chen M, Wang Y, Yang X. How Did Small Business Respond to Unexpected Shocks? Evidence from a Natural Experiment in China. Evidence from a Natural Experiment in China (March 26, 2022). 2022.

27.Huang D. How Effective Is Social Distancing? SSRN Electronic Journal. 2020(6186).

32.Yang Z, Chen Y, Huang Z. Research on Influencing Factors and Transmission Channels of Systemic Risks under International Shocks.Economic Research Journal.2023,58(01):90-106.

37.Chen M, Huang D, Wu B. Interlocking Directorates and Firm Performance: Evidence from China. Available at SSRN 4005022. 2022.

39.Chen M, Wang Y, Wu B, Huang D. Dynamic Analyses of Contagion Risk and Module Evolution on the SSE A-Shares Market Based on Minimum Information Entropy. Entropy (Basel). 2021;23(4):434.

40.Chen M, Li N, Zheng L, Huang D, Wu B. Dynamic correlation of market connectivity, risk spillover and abnormal volatility in stock price. Physica A. 2022;587:126506.

41.Wu B, Chen M, Huang D. Estimating Contagion Mechanism in Global Equity Market with Time-Zone Effects. Social Science Electronic Publishing.]

Comment#6. 2. Detailed Comments and Suggestions:

Comment#6.2.1- Please provide a more detailed explanation of the LASSO-VAR-DY framework and its advantages in analyzing the impact of temperature differences on systemic risk. This will help readers to better understand the methodology and its relevance to your research question.

The author's answer: We apologize for the lack of detail in the presentation of the framework, which caused problems in the editor's review process. We recognize your comments that the production of the LASSO-VAR-DY framework in the previous manuscript needed to be longer and more academic. We have refined the section "Research design" and adjusted the headings of each subsection appropriately. In the section "Construction of a pan-financial market volatility spillover network based on the LASSO-VAR-DY framework," we explain the advantages of the LASSO-VAR-DY framework in analyzing the impact of temperature differentials on pan-financial systemic risk. The risk spillover formula is supplemented in the "DY spillover index" section. At the same time, we have carefully analyzed and revised the model's overall presentation and refined this section in the new manuscript. We also provide detailed explanations of the specific settings of the model parameters in the empirical section to better serve the journal and readers in analyzing and replicating the results. The study design section of the uploaded manuscript has been revised and highlighted in red.

Comment#6.2.2- Consider conducting additional robustness checks to validate your findings. For example, you could explore alternative measures of temperature differences or systemic risk, or test the sensitivity of your results to different model specifications.

The author's answer: Thanks to the reviewers' suggestions for refining robustness, additional robustness tests were performed in this paper. The thesis uses six methods to test the robustness of the model. Firstly, in the regression analysis, the variables are deflated by 5% to consider the presence of extreme values in the data. Second, regarding the chance of sample time selection, the regression analysis is chosen to examine the robustness of the results for the sample from 2014 to 2021. Third, considering the effect of variable selection, the monthly average of temperature 2 meters above the surface (Temperature_2m) from the same database was selected to replace the core independent variable for the regression analysis. Fourth, considering the alternative measures of systematic risk of the explanatory variables, the choice of replacement of the explanatory variables selected the centrality of network topology features. This means that the Degree variable replaced the centrality of feature vectors for regression. Fifth, considering the sensitivity of the test results to different model scales, this paper resets the thresholds of network filtering. It selects 0.3 and 0.5 quartile thresholds to extract adequate information on the network obtained by Pearson's correlation coefficient. These six robustness tests, which consider alternative measures of temperature difference and systematic risk, are combined with the reviewers' suggestions to assess the sensitivity of the model specification to allow a robust demonstration of the empirical results.

Comment#6.2.3- In your discussion of the results, please provide more context on the potential mechanisms through which temperature differences may affect systemic risk. This will help to strengthen the theoretical underpinnings of your findings and provide a more comprehensive understanding of the relationship between climate change and financial stability.

The author's answer: Thanks to the reviewer's suggestion that the previous manuscript's results discussion section lacked background information on the potential mechanisms of temperature difference on pan-financial systemic risk, I have refined the relevant theoretical background in the empirical results discussion section of the latest version of the manuscript. In the area "Empirical analysis of the mechanisms influencing systemic risk in the pan-financial market," several subsections are devoted to the refinement of the background information on the mechanisms, the mechanism of temperature difference on systemic risk, the means of higher and colder temperature on systemic risk in the pan-financial market, and the theoretical mechanism of temperature difference on systemic risk in the subsector. In the section "Identifying potential pathways for the transmission of the temperature difference and systemic risk in the pan-financial market," the relevant theoretical background has also been added, and further refinements have been made for the contagion path of the minimum spanning tree and the dynamic evolutionary mechanism of temperature difference on systemic risk. These modifications are marked in red font in the trace manuscript..

Comment#6.2.4- It would be helpful to include a discussion of the policy implications of your findings. For example, what do your results suggest about the potential role of climate change mitigation policies in reducing systemic risk in the financial sector?

The author's answer: We apologize for the incomplete consideration of the policy proposals. We have made relevant changes in the Conclusions and Suggestions section of the manuscript. We discuss the implications of the findings of this study for policymakers, firm operators, and investors and make recommendations specific to each subject. The details are :[ First, promote cooperation among firms in pan-financial markets....... Second, it strengthens investors' ability to assess and manage climate risk....... Third, policymakers should consider climate change factors.......Fourthly, Governments should expand the scenarios for monetary policy and macroprudential regulation.......].

We apologize for the poor language of our manuscript. We worked on the manuscript for a long time, and the repeated addition and removal of sentences and sections led to poor readability. We have now worked on language and readability and have also involved native English speakers for language corrections. We hope that the flow and language level have been substantially improved.

Overall, these are just the revisions we have made. If the academic editors have found other problems in the literature review or think that any part of the revision of this paper still needs to be improved, please write to us again. We can do further communication to look forward to our cooperation more pleasant and our article more perfect. We will strive to publish this paper in PLOS ONE as soon as possible.

Thank you very much for your attention and time. I look forward to hearing from you.

Yours sincerely,

Kai-wei Jia

22 November 2023

School of Business Administration, Liaoning Technical University, Huludao, Liaoning,China

Te1:86+15104295228

E-mail:jiakaiwei@lntu.edu.cn

---

## [Decision Letter · Decision Letter 2]

10 Jan 2024

PONE-D-23-31011R2Temperature difference and systemic risk:

Evidence from LASSO-VAR-DY based on China's pan-financial marketPLOS ONE

Dear Dr. Jia,

Thank you for submitting your manuscript to PLOS ONE. After careful consideration, we feel that it has merit but does not fully meet PLOS ONE’s publication criteria as it currently stands. Therefore, we invite you to submit a revised version of the manuscript that addresses the points raised during the review process.

We look forward to receiving your revised manuscript.

Kind regards,

Islam Abdeljawad

Academic Editor

PLOS ONE

Journal Requirements:

Comments from PLOS Editorial Office: As mentioned in a prior email, in the previous rounds of review we noted that one or more reviewers recommended you cite specific previously published works. As always, we recommend that you please review and evaluate the requested works to determine whether they are relevant and should be cited. It is not a requirement to cite these works. We appreciate your attention to this request.

Reviewers' comments:

Reviewer's Responses to Questions

**Comments to the Author**

1. If the authors have adequately addressed your comments raised in a previous round of review and you feel that this manuscript is now acceptable for publication, you may indicate that here to bypass the “Comments to the Author” section, enter your conflict of interest statement in the “Confidential to Editor” section, and submit your "Accept" recommendation.

Reviewer #1: All comments have been addressed

Reviewer #2: (No Response)

2. Is the manuscript technically sound, and do the data support the conclusions?

Reviewer #1: Yes

Reviewer #2: Partly

3. Has the statistical analysis been performed appropriately and rigorously? 

Reviewer #1: Yes

Reviewer #2: No

4. Have the authors made all data underlying the findings in their manuscript fully available?

Reviewer #1: Yes

Reviewer #2: Yes

5. Is the manuscript presented in an intelligible fashion and written in standard English?

Reviewer #1: Yes

Reviewer #2: Yes

6. Review Comments to the Author

Reviewer #1: (No Response)

Reviewer #2: The topic of the article is interesting. However, the overall coherence of the text as well as the research conclusions are questionable. In addition, there are a number of stylistic errors throughout the article.

General remarks

1. The note is about causality. In his work, the author very often uses the term "impact" when he uses only regression-type methodology. The correct solution is to use methods based on the concepts of impact effects. The use of panel econometrics methods is a good substitute choice in the absence of counterfactual analyses. It is best to use the terms "relationship", "relationship", "association".

2. The note concerns the statistical significance of variables in regression models, to which the author attaches great importance and (also) on their basis verifies his hypotheses. In empirical studies, it often happens that statistical significance is not that important. I don't think it's defective. The thing fits into the current debate of scientists from various fields (excessive faith in p-value).

3. A flowchart should be added to the article to show the research methodology. Applied models, e.g. FE without any diagnostics. In addition, it is worth considering dynamic processes, i.e. the use of a model, e.g. GMM (The role of trade credit in business operations, Argumenta Oeconomica 2 (37), 189-231).

Additionally, in one FE regression you cannot use time variables (effect of time) and additional macroeconomic variables such as CPI, RATE, etc. The time effect includes all macro variables anyway.

Please see: Climate Risk with Particular Emphasis on the Relationship with Credit-Risk Assessment: What We Learn from Poland, Energies 14 (23), 8070

7. PLOS authors have the option to publish the peer review history of their article (what does this mean?). If published, this will include your full peer review and any attached files.

Reviewer #1: No

Reviewer #2: No

---

## [Author Response · Author response to Decision Letter 2]

13 Feb 2024

Response to PLOS ONE

To: "PLOS ONE" plosone@plos.org

From: "Kaiwei Jia " jiakaiwei@lntu.edu.cn

Subject: Temperature difference and systemic risk:

Evidence from LASSO-VAR-DY based on China's pan-financial market

Dear PLOS ONE :

Thank you very much for your comments and professional advice. These opinions help to improve the professionalism and standardization of my article. Based on your suggestion and requirements for the PLOS ONE journal, I have made corrected modifications to the manuscript. My changes to the manuscript are as follows: I have uploaded my manuscript and figures. Changes have been red-flagged in the Revised Manuscript with Track Changes file. The specific modifications are as follows:

Comment#1.General remarks :1. The note is about causality. In his work, the author very often uses the term "impact" when he uses only regression-type methodology. The correct solution is to use methods based on the concepts of impact effects. The use of panel econometrics methods is a good substitute choice in the absence of counterfactual analyses. It is best to use the terms "relationship", "relationship", "association".

The author's answer#1:First of all, I am very grateful to the reviewer for his suggestions regarding the use of regression methods, causality, and the term impact in this paper. I have carefully considered the term "relationship," which is indeed an excellent alternative to using panel econometric methods in the absence of counterfactual analysis. The term "impact" in this paper draws on relevant papers in the area of climate change and financial stability.

[reference19.Chabot M, Bertrand JL. Climate risks and financial stability: Evidence from the European financial system. J Financial Stab. 2023;69:101190.]. In this paper, the author estimates the influence of physical and transition risks on the European financial system through bank-level and system-wide measures of financial stability. They find that Scope 3 greenhouse gas emissions and chronic and acute climate risks negatively affect financial stability at both the financial institution and system levels. Temperature anomalies, heat waves, wildfires, and droughts are among the most significant risks.”Also, in this paper, the author points out that the influence of climate risks on financial stability is carried out using panel data models. A recent literature review on statistical approaches applied to climate and economics showed that the use of panel data is highly relevant when the research question aims to understand the response of a system as a whole to climate change.

[reference46.Song XN, Fang T. Temperature shocks and bank systemic risk: Evidence from China. Finance Research Letters. 2023;51.https://doi.org/10.1016/j.frl.2022.103447 ]. In this paper, the authors investigate the impact of temperature shocks on the systemic risk of Chinese listed banks, which is measured by a new nonlinear tail-event driven network (TENET) conditional value-at-risk (CoVaR) under expectation. We find that higher temperatures significantly increase the bank's systemic risk, and the impact of temperature shocks is significantly more significant during colder periods.

The areas of climate risk and financial stability examined in this paper are consistent with these papers, and both use panel regressions to examine the impact relationship between the two. Thus, the paper draws on the term "impact" used by these scholars in the discussion section of the main text.

Comment#2.General remarks :2. The note concerns the statistical significance of variables in regression models, to which the author attaches great importance and (also) on their basis verifies his hypotheses. In empirical studies, it often happens that statistical significance is not that important. I don't think it's defective. The thing fits into the current debate of scientists from various fields (excessive faith in p-value).

The author's answer#2:Thank you for the advice you have given. I also agree with you that empirical research should still be based on theoretical analysis. After all, the results of the empirical analysis are only support for the theoretical analysis. Thus, my selection of the topic is established on the basis of the theory rather than relying solely on the statistical significance. At the same time, the empirical results in this paper are all based on the software operation directly derived, accurate, and reliable. However, we also refer to the authors' viewpoints and make appropriate standardization and modifications for the expression of statements in the article.

Comment#3.General remarks :3. A flowchart should be added to the article to show the research methodology. Applied models, e.g. FE without any diagnostics. In addition, it is worth considering dynamic processes, i.e. the use of a model, e.g. GMM (The role of trade credit in business operations, Argumenta Oeconomica 2 (37), 189-231).

Additionally, in one FE regression you cannot use time variables (effect of time) and additional macroeconomic variables such as CPI, RATE, etc. The time effect includes all macro variables anyway..

The author's answer#3:On the first point, thank you for your suggestion to make a flowchart of the research methodology and add it to the paper in order to make the presentation of the research methodology more intuitive.

Fig 1. Research Methods

On the second point, thank you for providing the suggestion to consider dynamic processes in this paper. Since in the panel two-way fixed effects sector, the panel examined in this paper is a static panel (which means that the lagged terms of the explanatory variables are not included in the explanatory variables). Observing the linear relationship between higher and colder temperature and pan-financial systemic risk but failing to observe the dynamic evolution of the three variables may result in an unclear and ill-defined chain of causal logic. Therefore, in the section "Results and analysis of the dynamic evolutionary patterns of temperature difference and the systemic risk in pan-financial market," we choose to construct a panel vector autoregression model (PVAR) to analyze the systemic risk in pan-financial market and conduct an impulse response (IRF) analysis, so as to more intuitively portray the trend of the impact of temperature difference and the systemic risk in pan-financial market, and to study the dynamic time-varying patterns of evolution of the temperature difference and the systemic risk in pan-financial market.

GMM estimation is not used in this paper for the following reasons:

For GMM estimation, it is mainly used as a method to solve the endogeneity problem, which arises for three main reasons:(i) Omission of variables: for example, when studying the influencing factors of wages, the explanatory variables include years of education, but personal competence is not included due to the limitations of the measurement of the indicator, thus resulting in the omission of the influencing factors. (ii) Measurement error of variables: for example, we are always accustomed to using students' test scores to measure IQ, but this refers to the bias. A person's IQ can not be expressed simply through grades; a person's grades will be affected by other uncertainties; this way of measurement is too simple and refers to error. (iii) Bidirectional causality: for example, years of education affect wages.

On the one hand, The core explanatory variable used in this paper is "temperature difference," and the explanatory variable is "systematic risk." The temperature difference is a strictly exogenous variable compared with systematic risk. Most importantly, the judgment of endogeneity should be based on theoretical analysis, and the use of a series of Stata verifications only plays an auxiliary role. We generally will not be entirely in accordance with the state test to determine the existence of endogeneity.

[Form for endogeneity test]

Table . Endogeneity test.

 Dependent variable: systemic risk of pan-financial market

Temp40 0.4955

Temp80 0.5937

Temp120 0.5539

Temp_plus40 0.1697

Temp_minus40 0.1587

Temp_plus80 0.2272

Temp_minus80 0.1732

Temp_plus120 0.2051

Temp_minus120 0.1859

Note:Results are p-values for endogeneity tests.

The table shows the results of the endogeneity test for each variable as a core explanatory variable, which means that the original hypothesis is not rejected when the p-value is more significant than 0.05, proving that the model is not endogenous.

On the other hand, there is no bidirectional causality because only the temperature difference affects systemic risk, which does not affect changes in temperature. Finally, there may be an omitted variable problem, which is also considered in this paper, which uses a two-way fixed-effects model, with individual-fixed effects controlling for the inherent characteristics of firms that do not change over time and time-fixed effects controlling for the time trend of systemic risk. 

On the third point, thank you for your suggestion that "Additionally, in one FE regression, you cannot use time variables (effect of time) and additional macroeconomic variables such as CPI, RATE, etc. The time effect includes all macro variables anyway."We have reconsidered this issue and responded to it. In the first column of the robustness test in Table 6, re-doing the two-way fixed effects after removing the macro control variables. The results show that the effects of the variables are consistent with the original results. However, the goodness of fit is smaller than the original results, but this does not affect the robustness of the results. To this, we would like to give the following explanation. 

Analyzing empirical papers from an econometric writing perspective, this type of research methodology is also evidenced in many journal papers. For example, [1]Wang D, Wang Z, Cai W, et al. Digital inclusive finance, higher education expansion and regional carbon emissions:evidence from China[J]. International Review of Economics & Finance, 2024, 89: 1091-1101.[2]Chen Z, Huang W, Zheng X. The decline in energy intensity: does financial development matter?[J]. Energy Policy, 2019, 134: 110945.

And references cited in my paper [reference54.Aevoae GM, Andrieș AM, Ongena S, Sprincean N. ESG and systemic risk. Appl Econ. 2022;55(27):3085–109.https://doi.org/10.1080/00036846.2022.2108752.]. In this paper, the author used the control variables at the banking system level (Bank Concentration), country-level characteristics (Real GDP Growth and Inflation), and the overall level of governance in a country. Using a panel model that accounts for individual and time effects, the author documents a beneficial impact of the ESG Combined Score and Governance pillar on banks’ contribution to system-wide distress, analyzing a panel of 367 publicly listed banks from 47 countries over the period 2007–2020. The findings stress the importance of integrating banks’ ESG disclosure into regulatory authorities’ supervisory mechanisms as qualitative information.

Analyzed from a theoretical perspective, macroeconomic control variables and time effects in two-way fixed effects have different effects. Macroeconomic control variables are historical data, such as CPI, GDP, China Real Estate Climate Index, etc., and each company's stock volatility is closely related to a certain extent. It will have an impact on the role that must be addressed. At the same time, the historical data used for macro control variables may be lagged and noisy, which means that some unexpected events cannot be captured instantly. For example, in the 2008 economic crisis, real estate collapsed. However, macroeconomic variables were performing well at the time. When Lehman Brothers collapsed, the macroeconomic bank balance sheets in the United States could not reflect this sudden event, which means that time-fixed effects need to be incorporated to overcome these shocks to some extent.

Thus, this regression approach of incorporating two-way fixed effects of micro-level and macro-level control variables is also the result of our synthesis from both theoretical analysis and empirical measurement perspectives.

In addition to this, this paper also reorganizes the literature review section, adding to it the latest literature related to this topic.

In summary, here is my explanation for the fund change. If you have found other problems in the article or think that any part of the revision of this paper still needs to be improved, please write to us again. We can do further communication to look forward to our cooperation being more pleasant and our article being more perfect. We will strive to publish this paper in PLOS ONE as soon as possible.

Thank you very much for your attention and time. I look forward to hearing from you.

Yours sincerely,

Kai-wei Jia

13 February 2024

School of Business Administration, Liaoning Technical University, Huludao, Liaoning,China

Te1:86+15104295228

E-mail:jiakaiwei@lntu.edu.cn

---

## [Decision Letter · Decision Letter 3]

28 Feb 2024

Temperature difference and systemic risk:

Evidence from LASSO-VAR-DY based on China's pan-financial market

PONE-D-23-31011R3

Dear Dr. Jia,

We’re pleased to inform you that your manuscript has been judged scientifically suitable for publication and will be formally accepted for publication once it meets all outstanding technical requirements.

Kind regards,

Islam Abdeljawad

Academic Editor

PLOS ONE

Reviewers' comments:

Reviewer's Responses to Questions

**Comments to the Author**

1. If the authors have adequately addressed your comments raised in a previous round of review and you feel that this manuscript is now acceptable for publication, you may indicate that here to bypass the “Comments to the Author” section, enter your conflict of interest statement in the “Confidential to Editor” section, and submit your "Accept" recommendation.

Reviewer #2: (No Response)

2. Is the manuscript technically sound, and do the data support the conclusions?

Reviewer #2: Yes

3. Has the statistical analysis been performed appropriately and rigorously? 

Reviewer #2: Yes

4. Have the authors made all data underlying the findings in their manuscript fully available?

Reviewer #2: Yes

5. Is the manuscript presented in an intelligible fashion and written in standard English?

Reviewer #2: Yes

6. Review Comments to the Author

Reviewer #2: Dear Author,

Thank you very much. I accept actual vertion of your paper.

Kind regards

Reviever of this paper

7. PLOS authors have the option to publish the peer review history of their article (what does this mean?). If published, this will include your full peer review and any attached files.

Reviewer #2: No

---

## [Editor Report · Acceptance letter]

7 Mar 2024

PONE-D-23-31011R3 

PLOS ONE

Dear Dr. Jia, 

I'm pleased to inform you that your manuscript has been deemed suitable for publication in PLOS ONE. Congratulations! Your manuscript is now being handed over to our production team.

Kind regards, 

on behalf of

Dr. Islam Abdeljawad 

Academic Editor

PLOS ONE